# Design principles for energy transfer in the photosystem II supercomplex from kinetic transition networks

**Shiun-Jr Yang** [1,2,3], **David J. Wales** [4] ✉, **Esmae J. Woods** [4,5] & **Graham R. Fleming** [1,2,3] ✉

Photosystem II (PSII) has the unique ability to perform water-splitting. With light-harvesting complexes, it forms the PSII supercomplex (PSII-SC) which is a functional unit that can perform efficient energy conversion, as well as photoprotection, allowing photosynthetic organisms to adapt to the naturally fluctuating sunlight intensity. Achieving these functions requires a collaborative energy transfer network between all subunits of the PSII-SC. In this work, we perform kinetic analyses and characterise the energy landscape of the PSII-SC with a structure-based energy transfer model. With first passage time analyses and kinetic Monte Carlo simulations, we are able to map out the overall energy transfer network. We also investigate how energy transfer pathways are affected when individual protein complexes are removed from the network, revealing the functional roles of the subunits of the PSII-SC. In addition, we provide a quantitative description of the flat energy landscape of the PSII-SC. We show that it is a unique landscape that produces multiple kinetically relevant pathways, corresponding to a high pathway entropy. These design principles are crucial for balancing efficient energy conversion and photoprotection.

Photosystem II (PSII) produces oxygen by performing water-splitting[1]. Under ideal conditions, the charge separation processes in PSII can achieve a quantum efficiency near unity. However, an excessive amount of sunlight can saturate the electron transport capability, leading to triplet chlorophylls (Chls)[2], which, with the production of oxygen in PSII, causes the formation of reactive oxygen species (ROS) that are dangerous to living organisms[3]. Hence, PSII needs a robust photoprotection mechanism to maximise energy conversion efficiency and minimise photodamage, particularly for naturally fluctuating sunlight. Switching between efficient energy conversion and photoprotection in plants and algae involves numerous systems and complex processes that span a wide range of timescales[2]. From a microscopic point of view, the goal of these processes is to modulate the electronic energy transfer (EET) network to allow for the regulation of energy conversion. To understand the design principles required for efficient energy conversion, effective photoprotection, and the balance between these two requirements, we must investigate the energy transfer network in PSII.

PSII forms supercomplexes with light-harvesting complex II (LHCII). The $C_2S_2M_2$-type PSII-LHCII supercomplex (PSII-SC)[4] is the most common form in the thylakoid membrane under low light conditions (and sometimes under high light conditions)[5,6]. It is a dimeric system that consists of 24 pigment-protein complexes (Fig. 1), including a strongly bound LHCII (S-LHCII) and a moderately bound LHCII (M-LHCII) for each monomer. The multi-component design of the PSII-SC facilitates the repair process, as well as the control of

[1]Department of Chemistry, University of California, Berkeley, Berkeley 94720 CA, USA. [2]Molecular Biophysics and Integrated Bioimaging Division, Lawrence Berkeley National Laboratory, Berkeley 94720 CA, USA. [3]Kavli Energy Nanoscience Institute at Berkeley, Berkeley 94720 CA, USA. [4]Yusuf Hamied Department of Chemistry, University of Cambridge, Lensfield Road, Cambridge CB2 1EW, UK. [5]Cavendish Laboratory, Department of Physics, University of Cambridge, JJ Thomson Avenue, Cambridge CB3 0HE, UK. ✉e-mail: dw34@cam.ac.uk; grfleming@lbl.gov

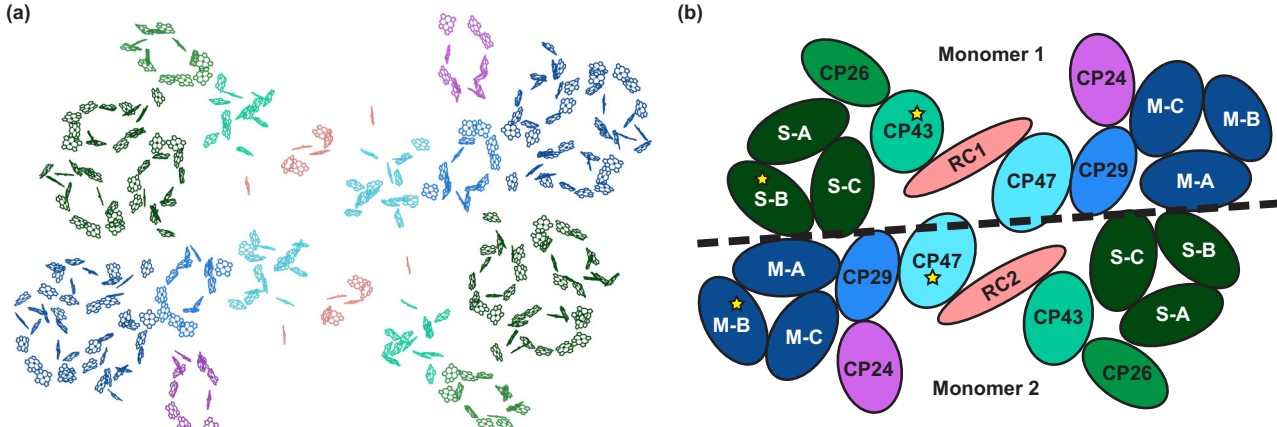

**Fig. 1 | Structure of the C$_2$S$_2$M$_2$-type PSII-SC. a** Pigment arrangement of the C$_2$S$_2$M$_2$-type PSII-SC adapted from the structure reported by Su et al. (PDB: 5XNL)[4]. **b** Labelling of the protein subunits. The colours of the subunits match those in (**a**), where the D1 subunits (CP43, CP26, and S-LHCII) are shown in green, the D2 subunits (CP47, CP29, CP24, and M-LHCII) are shown in blue and purple, and the RCs are shown in red. The black dashed line indicates the separation between the two PSII monomers, with the upper and lower monomers labelled as Monomer 1 and 2, respectively. The yellow stars represent the locations of the initial excitations discussed in the main text. RC: reaction centre. S-A: S-LHCII (A). S-B: S-LHCII (B). S-C: S-LHCII (C). M-A: M-LHCII (A). M-B: M-LHCII (B). M-C: M-LHCII (C).

antenna size. However, the design requires the subunits in the PSII-SC to work cooperatively to perform efficient EET, charge separation, and photoprotection. Numerous experimental and theoretical studies have been carried out to study the ultrafast EET dynamics in individual subunits or smaller complexes[7–16]. However, to reveal the design principles we must analyse the complete PSII-SC, i.e. how the interactions among the subunits encode the PSII functions. Fluorescence lifetime studies on the PSII-SC have shown that lifetimes of excitations increase with the size of antenna complexes[17–20]. These experiments provide a general understanding of the EET timescales involved in the PSII-SC, allowing for the construction of coarse-grained models. However, it remains difficult to account for the heterogeneity of EET pathways, particularly those involving different subunits. A model for the PSII-SC based on fluorescence lifetime fitting was proposed to deal with this issue[21]. However, the variations in connectivity between different subunits were treated implicitly and only the average effect of heterogeneity was obtained. An alternative approach for constructing a kinetic model is to simulate the EET dynamics from microscopic interactions. Bennett et al. proposed a structure-based model for the PSII-SC based on semi-empirical parameters obtained from smaller subunits and structure-defined interactions[22]. However, the structural information used in the simulations had limited resolution and new high-resolution structures have since become available[4].

A recent study employed a combined experimental and theoretical approach to investigate the EET pathways in the C$_2$S$_2$-type PSII-SC[23]. Two-dimensional electronic-vibrational (2DEV) spectroscopy provided enhanced resolution for capturing inter-protein EET. Simulations based on the state-of-the-art cryo-EM structure were performed to understand the EET network. Both the experiment and the simulations show that energy can flow in opposite directions depending on the location of the initial excitation, and more importantly, they also reveal the timescales involved in the EET pathways into and out of the PSII core. It was proposed that the bidirectional energy transfer on picosecond timescales is a key mechanism for balancing energy conversion efficiency and photoprotection. In particular, the ability for energy to flow in both directions, i.e. into and out of the PSII core, is encoded in the unique organisation of the energy landscape in the PSII-SC. Indeed, earlier studies[16,24,25] suggest that the PSII-SC has a relatively flat energy landscape, meaning that the energy levels of the pigments in all subunits are distributed in a similar range. This unique energy landscape is most likely required for PSII to support its ability to perform water-splitting, as shown by the observed bidirectional energy

transfer. However, understanding what other kinetic features the PSII energy landscape supports, and how these kinetic behaviours connect to the PSII function, requires further investigation of the overall EET network.

Here, we show how the design principles of the PSII-SC can be diagnosed from computational experiments on the underlying kinetic transition network[26–28]. Specifically, the roles of the PSII-SC subunits are explored by considering mutant networks where specific states are deleted in the kinetic analysis. In addition, the EET properties are encoded in the corresponding energy landscape, which can be directly visualised by translating the rates into free energies for the corresponding subunits and the transition states that connect them[29,30]. A similar approach has recently been applied to investigate polaritonic rate suppression[31].

## Results

We construct a kinetic model for the EET network in the C$_2$S$_2$M$_2$-type PSII-SC based on the method used by Bennett et al.[22] and Leonardo et al.[23] (see Methods for more details). We translate this kinetic transition network into the corresponding free energy landscape of the C$_2$S$_2$M$_2$-type PSII-SC, and investigate the dynamics using kinetic Monte Carlo (kMC) simulations[32–34]. A detailed description of these methods is given in the SI. This approach provides a way to perform single-trajectory analysis, which captures the inhomogeneity that cannot be reflected through averaged kinetic behaviour. To systematically characterise the EET network we have analysed first passage time (FPT) distributions and corresponding kMC trajectories for energy transfer. To facilitate the discussion, we label the upper and lower monomers as Monomer 1 and 2, respectively (Fig. 1). We focus on the cases in which the initial excitations, indicated by the yellow stars in Fig. 1b, are located in (i) CP43, (ii) S-LHCII (B), (iii) CP47, and (iv) M-LHCII (B). All initial excitations are placed in the left half of the PSII-SC, which includes the D1 subunits of Monomer 1 (CP43, CP26, and S-LHCII) and the D2 subunits of Monomer 2 (CP47, CP29, CP24, and M-LHCII). The left and right halves of the PSII-SC dimer exhibit almost identical dynamics, as expected from the approximate symmetry.

### First Passage Time Analysis

The FPT is the time it takes to reach a final state(s) for the first time from a defined initial state or distribution (Fig. 2a). In the context of EET in the PSII-SC, the final states we consider for FPT analysis are naturally the charge transfer species in the RCs, where most

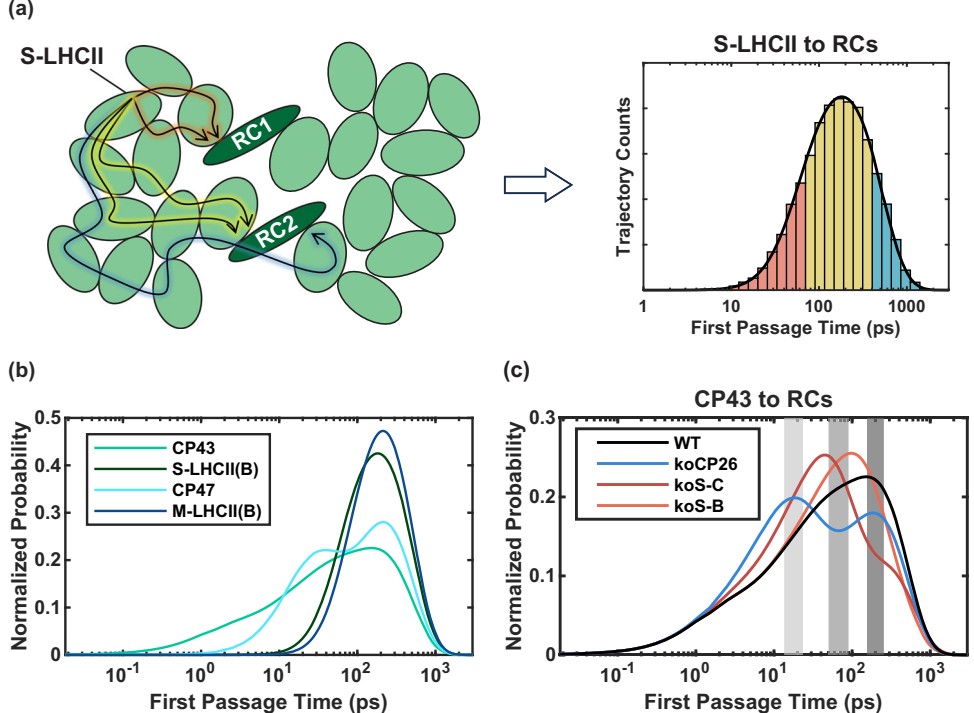

**Fig. 2 | First passage time analysis. a** Schematic representation of how the FPT distribution from S-LHCII to the RCs can be obtained from kMC trajectories. Short/medium/long trajectories, coloured in red/yellow/blue, contribute to the corresponding shaded area in the FPT distribution. **b** First passage time (FPT) distributions for the pathways from the initial excitation locations in Fig. 1b to the RCs in the WT PSII-SC. **c** FPT distributions for the WT and selected mutants of PSII-SC starting from excitation in CP43. The final state can be either RC. The FPT distributions are normalised according to the area under the curves. The shaded areas (light grey to dark grey) correspond to the FPT ranges discussed in the main text (see Dwell Time Distribution Analysis and Fig. 3). Source data files are available at https://doi.org/10.5281/zenodo.13346121.

excitations end up. Figure 2a shows a schematic representation of how an FPT distribution can be constructed based on kMC trajectories. The FPT of an EET pathway from the initial excitation location (S-LHCII in the example) to the RCs is exactly the duration (timescale) of the kMC trajectory. The FPT distribution therefore reflects the characteristic features of the EET pathways from S-LHCII to the RCs. The distributions of the FPT to either of the RCs were calculated using both the analytical solution of the master equation using eigendecomposition of the transition matrix, and kMC simulations for different initial excitation conditions. The analytical FPT distributions are normalised so that the integrated probability over the entire FPT range is unity. The results from the two methods are consistent (Suppl. Figures 1–4). First, we note that all the FPT distributions can span several orders of magnitude. For the WT (Fig. 2b), initial excitations in CP43 and CP47 can reach the RCs much faster than excitations in S-LHCII (B) and M-LHCII (B). This result is expected considering the difference in spatial proximity between these subunits and the RCs. Interestingly, the FPT distributions of the initial excitations in CP43 and CP47 clearly have multiple peaks. This structure is caused by the bidirectionality of EET, which allows some of the excitations to escape the PSII core at short times. In contrast, the FPT distributions of the initial excitations in the LHCII monomers exhibit a single peak, which is a result of the strong connectivity between the peripheral antennae. These features of the EET dynamics in different initial excitation conditions will be discussed in detail in the following sections and the SI. In addition to the differences originating from spatial proximity, the simulations also show a clear difference between initial excitations in the D1 antennae and in the D2 antennae. The FPT distribution of initial excitations in CP43 exhibits a shoulder for shorter FPTs (peaking at around 1 ps), which is absent in CP47. Furthermore, initial excitations in S-LHCII (B) lead to slightly shorter FPTs than those in M-LHCII (B). This difference is direct evidence of faster EET from CP43 to the RC than from CP47.

This phenomenon has been discussed by Raszewski and Renger[14], who proposed that the difference results from the location of the lower energy states, which are closer to the RC in CP43 than in CP47. It is important to note that in Ref. 14, the distribution function does not feature the short transfer timescale (~1 ps) between CP43 and the RC because it is based on thermally averaged timescales of different inhomogeneous realisations, whereas the FPT distribution reveals timescales involved in different pathways. The difference arises from the fact that the averaged timescales do not highlight individual pathways. In fact, there is no major discrepancy between our results and those reported in Ref. 14, as discussed in more detail in the SI.

The removal of protein subunits causes systematic changes to the FPT distributions and the mean first passage times (MFPTs) (Fig. 2c), and the changes are very different for different knockout mutants. These results indicate that each protein subunit has its own function in the EET network. For example, Fig. 2c shows the analytical FPT distributions of the WT and selected mutants for initial excitations in CP43. In general, the FPT distributions shift to shorter timescales when the selected subunits are removed. CP26 knockout (koCP26) shows the most obvious difference from the WT—the FPT distribution is shifted more than any other mutants and two distinct peaks are observed. The FPT distributions of S-LHCII (B) and (C) knockouts (koS-B, koS-C, respectively) also shift to shorter timescales, but maintain just one clear peak with shoulders of varying amplitude. Discussion of the other excitation conditions can be found in the SI.

**Dwell time distribution analysis**

To relate the structures of the FPT distributions to EET pathways, we analyse the dwell time distributions, i.e. the amount of time spent in each subunit, averaged from the kMC trajectories for specific time ranges (Fig. 2c, grey areas). The dwell time distributions reveal which subunits the excitations visit and primarily reside in before reaching

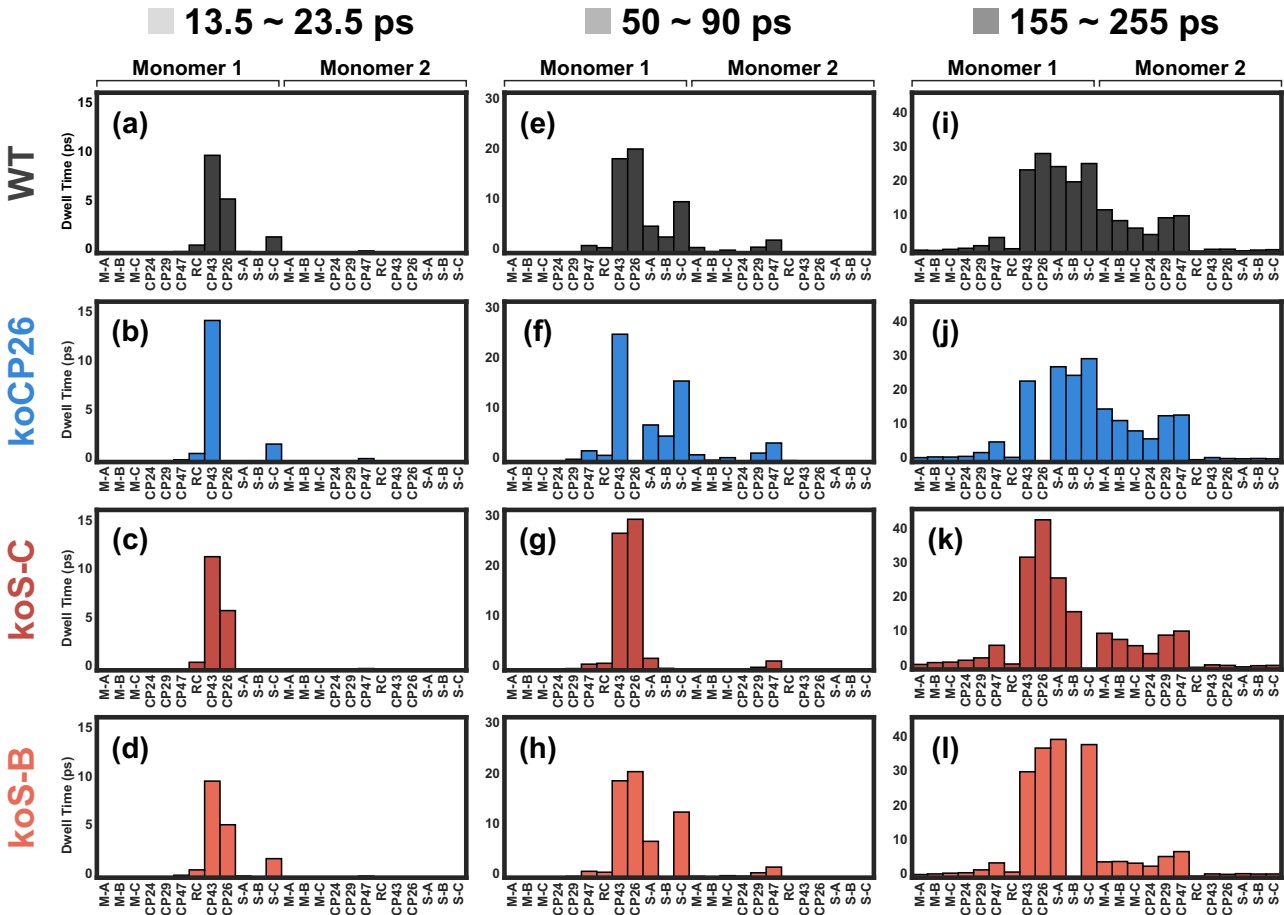

**Fig. 3 | Dwell time distribution for CP43 initial excitation.** Dwell time distributions extracted from the trajectories of initial excitations in CP43 in the FPT range of (**a–d**) 13.5 to 23.5 ps (Fig. 2c, light grey), (**e–h**) 50 to 90 ps (Fig. 2c, grey), (**i–l**) 155 to 255 ps (Fig. 2c, dark grey). The distributions from top to bottom are for the WT, koCP26, koS-C, and koS-B, respectively. In each panel, categories on the left are the subunits in Monomer 1 and categories on the right are those in Monomer 2. Source data files are available at https://doi.org/10.5281/zenodo.13346121.

the RCs. Here, we continue to focus on the cases in which initial excitations are in CP43. Discussion of the other excitation conditions can also be found in the SI.

First, we look at the trajectories with an FPT between 13.5 and 23.5 ps (Fig. 3a–d). The WT dwell time distribution in this time range shows that energy primarily travels from CP43 toward CP26 and S-LHCII (C). As a consequence, the removal of these two subunits results in more trajectories with shorter FPTs (Fig. 2c, blue and dark red) compared to the WT (Fig. 2c, black) as the pathways out of the PSII core are partially blocked and the chances of entering RC 1 increase. The dwell time distribution also shows that the visits to CP26 are much more frequent than visits to S-LHCII (C), indicating that the connectivity between CP43 and CP26 is much stronger than that between CP43 and S-LHCII (C) for the shorter trajectories in this range (13.5 to 23.5 ps). Since RC 1, CP43, CP26 and S-LHCII (C) are the primary subunits involved in the kMC trajectories on this timescale, the absence of other subunits does not result in any significant changes to the FPT and the dwell time distributions from the WT.

Next, we look at the trajectories with an FPT between 50 and 95 ps. For the WT, the FPT probability is higher in this FPT range (Fig. 2c, grey) than for shorter trajectories (Fig. 2c, light grey). The corresponding dwell time distributions (Fig. 3a,e) reveal that the major difference between these two timescales is that visits to other antenna subunits in Monomer 1 and the D2 subunits in Monomer 2 are much more significant on this timescale than on the shorter timescale (13.5 to 23.5 ps). Therefore, the higher probability in this FPT range indicates

strong connectivity between the two monomers in the antenna system, which allows transfer from one monomer to the other to occur more often than not upon excitations in CP43. Interestingly, the absence of CP26 does not significantly alter the dwell time distribution despite causing a decrease in probability in the FPT distribution (Fig. 2c, blue) on this timescale. This result indicates that the EET pathways to the other peripheral antennae do not necessarily involve CP26. However, without CP26, the transfer back to RC 1 will be partially blocked, which results in the decreased FPT distribution. This interpretation is also supported by the relatively higher probability at longer FPT (note that the peak is not higher than the probability distribution of the WT due to a significant number of initial pathways out of the core being blocked, while the total area is normalised). For koS-C, which lacks S-LHCII (C), the dwell time is significantly decreased for S-LHCII and is eliminated almost completely for M-LHCII. This observation indicates that S-LHCII (C) is crucial for transferring energy into both LHCII trimers from CP43 on this timescale, especially to M-LHCII. The absence of S-LHCII (B) also reduces the dwell time in the D2 subunits of Monomer 2, but is less disruptive than the absence of S-LHCII (C), which suggests that the most important pathways for transfer from CP43 to LHCII in this time range involve S-LHCII (C).

Finally, we examine the trajectories with an FPT between 155 and 255 ps. The dwell time distribution (Fig. 3i–l) clearly shows that energy can travel throughout the left half of the PSII-SC on this timescale, with some trajectories terminating in RC 2. On this timescale, some trajectories can even explore the right half of the PSII-SC, as shown by the

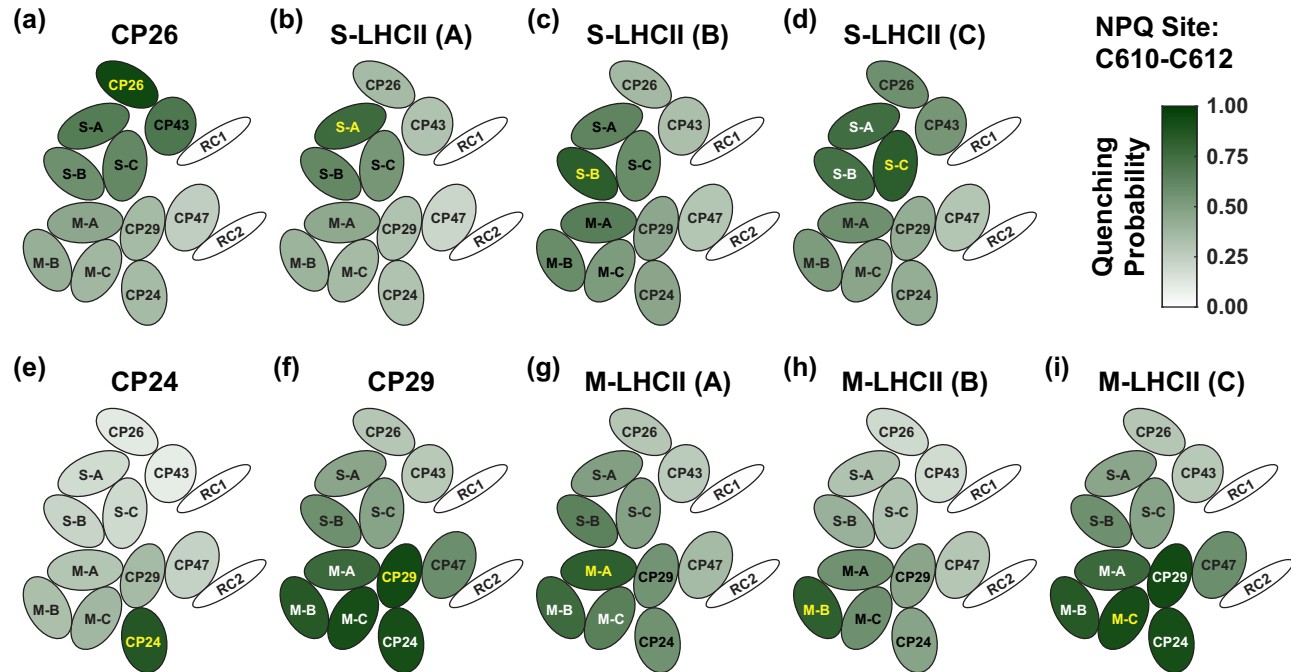

**Fig. 4 | Simulation of NPQ with different quenching sites.** Probability of initial excitations in each subunit being quenched when the quencher is placed in (**a**) CP26, (**b**) S-LHCII (A) (S-A), (**c**) S-LHCII (B) (S-B), (**d**) S-LHCII (C) (S-C), (**e**) CP24, (**f**) CP29, (**g**) M-LHCII (A) (M-A), (**h**) M-LHCII (B) (M-B), and (**i**) M-LHCII (C) (M-C). The subunit where a quencher is placed is shown in yellow. The darker green a subunit is, the more likely the initial excitations in that subunit are quenched. Only subunits on the left side of the PSII-SC are shown (see Fig. 1). Source data files are available at https://doi.org/10.5281/zenodo.13346121.

nonzero dwell time at either end of the plots. The removal of CP26 has almost no effect on the dwell time distribution in this FPT range. This result shows that, while the absence of CP26 blocks some pathways leading to RC 1, there are alternative pathways or even pathways leading to RC 2 that are more relevant on longer timescales. Similarly, the removal of S-LHCII (C) completely blocks the transfer to M-LHCII for the trajectories with an FPT between 50 and 90 ps, but energy can eventually reach M-LHCII in this longer FPT range. This observation also indicates the presence of alternative pathways, most likely through S-LHCII (B) and M-LHCII (A). The most distinct dwell time distribution from the WT is found for koS-B. The absence of S-LHCII (B) leads to much shorter dwell times in M-LHCII subunits, suggesting that S-LHCII (B) is important for energy transfer to M-LHCII.

## Simulation of non-photochemical quenching

To further understand the role of each subunit in photoprotection, we also performed a simulation of EET when non-photochemical quenching (NPQ) is active. In plants, excessive illumination will introduce a pH gradient across the thylakoid membrane. This gradient activates a pH-sensing protein and the xanthophyll conversion cycle[35]. The xanthophyll produced under highlight conditions, namely zeaxanthin, can act as a quencher, accepting excitation energy from the Chls and removing it rapidly via dissipative pathways[2,3]. To simulate energy quenching, an additional component is added to the rate matrix as a sink. While the exact mechanism of NPQ involves many processes, here we treat the quenching phenomenologically by focusing on how active quenching affects the EET dynamics within the PSII-SC. It is generally accepted that quenching in PSII occurs in the peripheral antennae. However, the specific subunits involved in quenching (and whether there is one or more) is still an open question. To understand how different quencher locations result in different photoprotection abilities, we performed NPQ simulations by placing a single quencher in one peripheral antenna subunit in each monomer at a time and investigated all the possibilities for peripheral antenna subunits. Here, we also focus on the cases where initial excitations are

placed on the left side of the PSII-SC (Fig. 1). Figure 4 shows the probability of the excitation first reaching the quencher before reaching either RC. The green scale of each subunit indicates the probability of initial excitations in the subunit being quenched. In general, the higher the probability of reaching a quencher first, the better photoprotection is achieved. It is clear that different quencher locations lead to different photoprotection abilities.

Overall, the best photoprotection ability is achieved by placing the quencher in CP29 and M-LHCII (C). We note that these two distributions are practically identical because the Chls in these two subunits that transfer energy to the quencher belong in the same domain, indicating that EET between the two sets of Chls is fast and that they can quickly reach a local equilibrium (see Methods for more details)[22]. In other words, placing the quencher in CP29 has the same effect as placing it in M-LHCII (C) because EET between the two sets of Chls is much faster than EET out from them. The distribution shows that initial excitations in the D2 peripheral antennae are mostly quenched (> 80%), while the quenching efficiency for initial excitations in the D1 antennae is not as high (<50%) because energy needs to pass through CP29 before reaching the RC 2. Therefore, there is a high probability for energy to be quenched if the quencher is placed there. In contrast, for initial excitations in the subunits on the D1 side, the fast EET between CP43 and RC 1 allows energy to reach RC 1 without much chance of passing through CP29. Placing the quencher in M-LHCII (A) and M-LHCII (B) results in similar distributions, although the quenching probability is not as high as placing it in CP29 for the excitations on the D2 side. Placing the quencher in CP24 leads to the poorest photoprotection ability, because CP24, located at the edge of the PSII-SC, only plays a minor role in the overall EET network. In most scenarios, energy does not visit the minor antenna.

When the quencher is placed in the D1 peripheral antennae, the photoprotection ability is similar for the excitations on the D1 and the D2 side, except for CP26. This result differs from quenchers in the D2 peripheral antennae, and the reason is that S-LHCII, as a complete trimer, is well connected with both M-LHCII and CP43. However, none

**Table 1 | Summary of the functional role of the PSII-SC subunits**

| Subunit | Functional Role |
|---|---|
| **LHCII** | Collecting photons and distributing energy to both RCs; Photoprotection |
| S-B, M-A | Connecting the two monomers |
| S-C | Connecting CP43 and CP29 |
| **Minor Antennae** | Collecting photons |
| CP29, CP26 | Good candidates for photoprotection |
| CP24 | Structural stabilisation |
| **Core Antennae** | Balancing efficiency and photoprotection |
| CP47 | Slow transfer to RC |
| CP43 | Fast transfer to RC |

of the S-LHCII monomers have exclusive EET pathways to either RC, i.e. alternative pathways exist almost everywhere on the D1 side. The result of this organisation is that the quencher placed in any of the S-LHCII monomers has the ability to quench the excitations on both the D1 and the D2 sides, but the quenching probability is not as high as when the quencher is placed in CP29. When the quencher is in CP26, the fast EET between CP43 and CP26 produces the most effective photoprotection of CP43. Since all excitations entering the RC 1 have to pass through CP43, which has a strong connection with CP26, the quencher in CP26 can also quench the excitation in other S-LHCII monomers.

### Roles of the PSII-SC Subunits

Based on the global kinetics analysis, the functional role of each subunit can be inferred. In this section, we discuss how the observed EET network allows the subunits of the PSII-SC to contribute to balancing efficiency and photoprotection. A summary is shown in Table 1.

### LHCII: Energy Collection and Distribution

LHCII trimers, binding both Chls $a$ and Chls $b$, are the major antennae for light harvesting. They are crucial for collecting sufficient photons for PSII to complete the multielectron redox cycle, especially under low light intensity, where the incidence of photons is relatively rare for each Chl. For the $C_2S_2M_2$-type PSII-SC, which is the dominant species among PSII complexes under low light conditions, both S-LHCII and M-LHCII should be able to transfer energy to both RCs in the PSII dimer in order to maintain high efficiency for each of them. As shown in the Results section, there are EET pathways connecting S-LHCII and M-LHCII, particularly between S-LHCII (B) and M-LHCII (A). These pathways are fast compared to the overall EET to the RC, allowing most excitations to travel to both monomers and enter either RC. This dynamical feature improves the ability of the antenna system to harvest sunlight for both RCs, which is crucial because statistically excitations do not always distribute evenly between the two LHCII complexes under low light conditions.

On the other hand, having strong connections between S-LHCII and M-LHCII also opens up many pathways that do not directly lead to the RCs. For example, initial excitations in S-LHCII (B) can reach the RCs in less than 30 to 50 ps for direct transfer, as discussed in the SI. However, direct transfer only contributes to a small number of trajectories. The majority of trajectories, which have an FPT between 135 and 235 ps (Fig. 2b), involve visits to both S-LHCII and M-LHCII. These pathways have many more steps than direct transfer, allowing excitations to explore the antenna system, which also contains quenchers for photoprotection. Their presence results in much longer timescales for the overall transfer to the RCs. For example, the MFPTs for initial excitations in S-LHCII (B) and M-LHCII (B) are 229 and 255 ps, respectively. Nevertheless, these timescales are still much shorter than nonradiative decay of Chl excited states, allowing excitations to reach the

RCs well before they are dissipated. In addition, there are also pathways between S-LHCII (C) and CP29; although the probability of going through them is lower than for routes connecting S-LHCII (B) and M-LHCII (A), they provide a faster path between the two PSII monomers.

Overall, the two LHCII in the $C_2S_2M_2$-type PSII-SC on each side are responsible, not only for increasing photon collection rates, but also for connecting the two PSII-SC monomers. This design allows both RCs to maintain high efficiency under low light conditions, and it also enhances photoprotection ability by increasing the probability of excitations visiting the quenchers. Furthermore, the time spent in the two LHCII is not long enough to prevent excitations from reaching the RC before energy is dissipated. Additional LHCII may cause the number of steps in the pathways to the RCs to increase nonlinearly, which can lead to much longer EET timescales.

### Minor antennae: good candidates for photoprotection

Despite the structural similarity, the three minor antennae present in the $C_2S_2M_2$-type PSII-SC have very different interactions with other subunits and therefore different working mechanisms. These differences arise from the alternative protein orientations and arrangements. CP29, located on the D2 side, is the only subunit connected to CP47. It mediates a fast route with S-LHCII (C) for transfer between the two monomers. It also connects CP47 and M-LHCII, allowing EET in both directions. Since all excitations on the D2 side have to pass through CP29 to enter or exit the PSII core, it is an ideal location for a quenching site. As Fig. 4f shows, the probability of excitations in the D2 subunits passing through CP29 is high, allowing a quencher within it to effectively quench the excitations when activated. In this case, the timescales do not have a strong effect on the functions of CP29 because excitations have to pass through it whether the transfer is fast or slow.

Unlike CP29, CP26 is not the only subunit connecting CP43 and S-LHCII. While CP26 is next to CP43 and S-LHCII (A), S-LHCII (C) is also in direct contact with CP43 (Fig. 1b), allowing direct transfer from S-LHCII (C) to CP43 without passing through CP26. Nevertheless, CP26 is still strongly connected to CP43. In fact, excitations in CP43 have a very high probability of visiting CP26. Because of the strong connection between CP26 and CP43, excitations in CP26 can also quickly return to CP43. The whole process of visiting both CP26 and CP43 and entering the RC can be as fast as 10 ps. The strong connection makes CP26 a good location for a quenching site, although for a different reason from CP29. For excitations entering the RC from CP43, there is a high chance it will be transferred to CP26 before reaching the RC. If the quencher in CP26 is activated, excitations are very likely to be quenched. If the quencher is inactive, excitations can still enter the RC on a short timescale to ensure efficient energy conversion. Furthermore, excitations in S-LHCII can also be transferred to CP26 through S-LHCII (A), but the reverse transfer is less likely to occur given the fast EET between CP26 and CP43. The preferential transfer in CP26 to CP43, rather than S-LHCII (A), introduces a directionality for excitations in S-LHCII (A) and S-LHCII (B), which improves the energy conversion efficiency on the D1 side.

Overall, both CP29 and CP26 are good candidates for a quenching site, even though the working mechanisms are predicted to be different. CP29 occupies a pivotal position in the EET network on the D2 side and can work on different timescales, depending on the subunits it interacts with. CP26 has a fast and strong connection with CP43, which allows a quick detour that facilitates the balance between efficient energy conversion and photoprotection. CP24 has only a minor effect on the overall EET network in the PSII-SC. It seems likely that its presence is more important for stabilisation of M-LHCII. We note that there is an ongoing debate on whether CP29 and CP26 actually perform energy quenching[36–42]. In particular, it has been suggested that CP26 does not have a direct or indirect effect on the energy

transfer of the PSII-SC, as its removal has only a slight effect on the fluorescence lifetime[18]. In addition, it has been shown that the removal of CP26 and CP29 does not have a significant effect on the NPQ activity measured based on fluorescence yields[43,44]. We stress that the analysis of fluorescence data on mutant studies is complex. First, fluorescence lifetime is only a measure of averaged kinetics instead of microscopic pathways. Due to the complexity of the system, different combinations of EET pathways can still lead to similar fluorescence lifetimes. Second, inhomogeneous excitation of the PSII-SC can undermine the effect of a single subunit in the overall EET network, which is averaged out by other pathways that do not involve the subunit. Most importantly, it is clear that deletion of the minor antenna complexes significantly perturbs the structure of the PSII-SC[19,37,39,45,46]. This effect alone could change fluorescence behaviour, making it difficult to interpret the specific functional roles of the removed complex based on fluorescence measurements in mutant studies. In contrast, the results and analysis in this work refer to the roles of these complexes within an intact $C_2S_2M_2$-type PSII-SC, a perspective that is not available from mutant studies. We are not able to comment on the roles of CP26 and CP29 that are detached from the PSII-SC[47]. Such a discussion is also not within the scope of the current work. Rather, we have performed kinetic analysis to show that, due to the different ways the minor antennae participate in the energy transfer network, they can have different abilities and different working mechanisms for maintaining efficiency and performing photoprotection.

### CP43 and CP47: regulation of efficiency and photoprotection

The core antennae, CP43 and CP47, are responsible for collecting sunlight and, more importantly, connecting the peripheral antenna system and the RC. They are capable of transferring energy in both directions, i.e. to the RC and to the peripheral antennae. Transferring energy from the core antennae to the peripheral antennae effectively allows excitations to visit quenching sites, and is, therefore, an important mechanism for photoprotection. Many studies have suggested that EET from the core antennae to the RC is the slowest step in the EET pathways of the PSII-SC due to the large distances between them[9,16,48–50]. Sometimes it is proposed to be the rate-determining step. The FPT distributions (Fig. 1c) show that direct transfer from CP43 to the RC occurs on a sub-picosecond to picosecond timescale, whereas for CP47 to RC it takes the order of 10 ps. These timescales are much shorter than the fluorescence lifetime (around 150 ps), indicating that EET from the core antennae to the RC is not the only step contributing to the lifetime. Indeed, the structure of the FPT distributions for initial excitations in CP43 and CP47 (Fig. 2b) clearly shows that pathways for direct transfer and those involving visits to the peripheral antenna have significantly different timescales. This result indicates that transfer from CP43 and CP47 to the RC is not a clearly defined rate-limiting step. Nevertheless, it is important that this step is slower, so that EET from core antennae to peripheral antennae can compete with it to allow for effective photoprotection. It is also important that this step is not so slow that it prevents excitations from reaching the RCs before they undergo dissipation. Therefore, CP43 and CP47 have a crucial role in balancing the rates of EET to peripheral antennae and to the RC, ensuring that the EET network meets the requirements for both efficient energy conversion and effective photoprotection.

Interestingly, the timescale of EET from CP43 to the RC is much shorter than from CP47 to the RC. This difference suggests that CP43 and CP47 may have slightly different functions, even though in general they are both functioning as bridges connecting peripheral antennae and the RC. CP43 allows faster EET to the RC, but the probability of directly transferring to the RC is low (Fig. 2b, light green). This scenario arises because the rate of transfer to the peripheral antennae is also fast, resulting in a high probability of excitations leaving the PSII core. In contrast, CP47 allows slower EET to the RC, but the probability of direct transfer is much higher (Fig. 2c, light blue). Overall, both CP43

and CP47 are able to balance energy conversion and photoprotection by transferring energy in both directions, but the timescales involved are very different. This difference leads to a slightly shorter MFPT for initial excitations in CP43 than for initial excitations in CP47 (129 vs 166 ps). Since the peripheral antennae provide a strong connection between the two monomers, the higher probability of EET from CP43 to the RC also allows excitations in peripheral antennae to preferentially enter the RC from the D1 side, which directly leads to the active branch. This asymmetry in the EET network suggests that CP43 has a more important role in efficient energy conversion and CP47 is more important for photoprotection, particularly in combination with CP29. However, whether the asymmetry is significant enough to cause a functional difference between CP43 and CP47 requires further investigation.

We note that CP47 does have an additional role in connecting the left and right side of the PSII-SC. This role is facilitated by the close proximity of the two CP47, which has been discussed in a study of the PSII core complex[51]. However, the transfer between the two CP47 is not as significant as EET from CP47 to CP29 or to the RC.

### Mechanisms of the PSII-SC EET Network

Since the unique ability of the PSII-SC to perform water-splitting increases the risk of photodamage, it has evolved a multi-component structure to overcome this survival challenge. This design allows the PSII-SC to be dismantled and reassembled, facilitating the repair process. It also supports photoprotection that involves the activation and deactivation of quenchers, which requires complex interactions with other proteins. In addition, it provides a way for the antenna size to be systematically controlled, responding to the sunlight condition. In particular, tuning the interplay between entropy and enthalpy, such as attaching/detaching antenna proteins or introducing energetic sinks, allows the EET work to be modulated to favour efficiency or photoprotection. Understanding the control of the EET network in the multi-component design is therefore crucial for revealing the design principles of the PSII-SC.

EET dynamics in the PSII-SC are controlled by the energetics of the Chls and their interactions. Structural information provides insight into the interactions, but understanding the energy landscape of the PSII-SC is essential for understanding the EET network. The transfer rates, although they obey detailed balance, are not determined by a barrier. However, we can visualise the corresponding kinetic transition network in terms of effective free energies corresponding to the states in the model, and the transition states that connect them. The purpose of this construction is to provide a comparison with molecular free energy landscapes, to provide insight into the evolutionary design of the PSII supercomplex and relate this design to the functions and constraints it must fulfil. The structure of the corresponding free energy landscape can be visualised directly by translating the rate matrix into a disconnectivity graph[52,53]. This graph provides a visualisation of the effective free energies of the substates, and the transition states that connect them, where the vertical axis corresponds to the free energy. Our construction produces a graph that faithfully reproduces the corresponding rates and equilibrium occupation probabilities[29,30] (Fig. 5). The branches of the graph terminate at effective free energies for the substates of the PSII-SC, which are the exciton states, and the relative free energies correspond to the equilibrium distribution for the corresponding rate matrix. We join the branches for individual minima when the free energy reaches the threshold corresponding to the highest transition state on the lowest energy path that connects them. Here, the rates between substates are translated into free energies for the effective transition states, so that the entries in the rate matrix are recovered from the corresponding free energy difference, as explained in the SI.

Once the branches merge they represent sets of minima that can all interconvert below the free energy threshold defined by the vertical

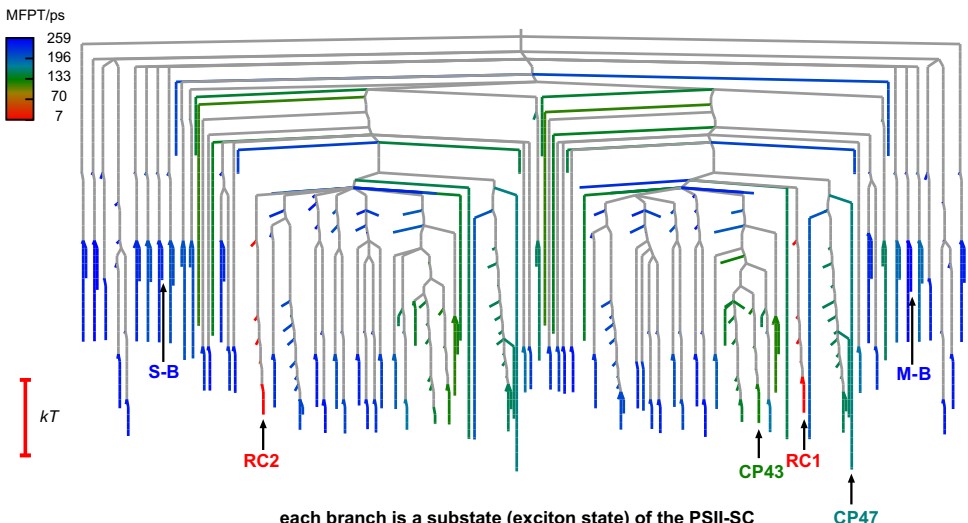

**Fig. 5 | Energy landscape of the PSII-SC.** Free energy disconnectivity graph[52,53] for the original WT PSII-SC kinetic transition network created using the disconnectionDPS program[78]. Each branch terminates at one of the exciton states of the PSII-SC, which is the basis for EET calculations. Free energy increases on the vertical axis. The branches that terminate at single substates are coloured according to the mean first passage time for energy transfer to either RC for that substate, as indicated in the key. The branches close to the RC are red in this colour scheme because they have the shortest MFPT values. The initial excitation locations (CP43, CP47, S-B, and M-B) and the final states (RC1 and RC2) discussed in the Results section are pointed out by the arrows and labelled accordingly. The approximate two-fold symmetry in the graph reflects the dimeric structure of the complex. Source data files are available at https://doi.org/10.5281/zenodo.13346121.

axis. At the top of the graph, all the substates can interconvert, and there is only one branch. If we think about the graph from the top down, the branches representing multiple substates split into disjoint sets when the free energy falls below the highest point on the lowest energy path that connects them. The ordering of substates on the horizontal axis is determined by how the sets split as the free energy decreases, providing a visualisation of the landscape organisation that quantitatively encodes the underlying rate matrix. The spacing between branches that split is chosen to accommodate the branches that will appear as the free energy decreases further, and the approximate dimeric symmetry of PSII emerges naturally from this construction.

Figure 5 illustrates how PSII-SC has evolved a landscape to support and balance efficient energy conversion and photoprotection. The components span a relatively narrow range of energy of around kT at the temperature corresponding to calculated rates. All the states are mutually accessible with no significant kinetic traps. Due to these features and the complexity of the system that enables multiple accessible pathways, the overall dynamics are largely independent of temperature. This structure is very different from the classes of landscape we have identified before[53], and represents a new motif, which reflects the unique functionality. Single funnel landscapes correspond to self-organising or 'structure-seeking' systems[53], which support kinetically convergent pathways[54] consistent with the concept of minimal frustration[55]. Encoding multiple functions, such as a molecular switch, requires a double-funnel[53,56–58] or multifunnel[59,60] landscape. In contrast, the landscapes characterised for structural glasses have a multitude of low-lying minima, separated by barriers that are large compared to the glass transition temperature[61–63], leading to broken ergodicity on cooling[64–66]. The PSII-SC landscape exhibits low-lying free energy minima in a narrow energy range, but it is not glassy because there are no kinetic traps. The closest analogue for this PSII-SC organisation is perhaps the loss function landscape characterised for neural networks with multiple hidden layers in the overfitting regime[67].

Our simulations further show that the relatively flat energy landscape has important consequences beyond the bidirectional EET between core antennae and peripheral antennae. For example, this organisation also facilitates bidirectional energy flow between S-LHCII

and M-LHCII, which allows both antennae to harvest sunlight for the RCs in the two monomers. It is important to understand how the relatively flat energy landscape controls the EET network, as the kinetics will be quite different from single- or multi-funnel landscapes. For a relatively flat landscape it is the pathway entropy, i.e. alternative kinetically relevant paths between states[63], that plays a key role in determining the kinetic behaviour. The probability distribution for the different pathways, encoded by the energy landscape and pathway entropy, must be an important factor that enables the PSII function. We surmise that this energetic structure is the evolutionary solution to the constraints of photoprotection and efficient energy conversion.

For the PSII-SC, the same initial excitation can reach the RCs through very different pathways. The associated timescales can span four orders of magnitude (Fig. 2b). The range of timescales associated with alternative relaxation paths on multifunnel landscapes with kinetic traps that induce rare events and broken ergodicity can be much larger[30,68]. These alternative pathways produce well separated peaks in the FPT distribution, corresponding to kinetic traps. In contrast, the PSII-SC landscape features distinct paths, and a distribution of timescales, but without trapping. In fact, our simulations show that the timescales for different pathways leading to the RCs are primarily determined by the number of steps (the dynamical activity[69,70]), which differs drastically from one trajectory to another. This result shows that there is no single step that can fully describe the kinetic behaviour of the system, and no particular transfer between one complex to another contributes to the majority of the lifetime. It is the probability of going through different pathways, long or short, that dictates the overall EET timescales. Because of the wide variety of pathways, the mean lifetime is not a useful description of the EET network in the PSII-SC. Therefore, it is necessary to understand the probability distribution of different pathways, i.e. the FPT distribution, in order to capture the heterogeneity of the EET network.

The pathway heterogeneity originating from the unique PSII-SC landscape leads to a question: How is the number of steps controlled? In a relatively flat energy landscape, excitations in different subunits do not result in a large energy difference. In other words, the enthalpy change for excitations to move from one subunit to another is not significant within the EET network until they reach the RC, where a

large drop in enthalpy occurs with charge separation to trap the energy. As a result, entropy plays an important role for EET in the PSII-SC, and the number of steps is controlled by the interplay between entropy evolution and enthalpy evolution. The distribution of the number of steps, highly correlated with the FPT distribution, is a deciding factor in the overall EET timescales. Therefore, it is very likely that the entropy evolution in the PSII-SC is connected to the timescales of the EET network and energy trapping, which encodes its ability to balance efficiency and photoprotection. While quantifying entropy evolution and relating it to PSII-SC function is not within the scope of this work, it is an area for future exploration that could bring deeper understanding of the design principles of the PSII-SC.

## Discussion

We have presented a detailed analysis of the electronic energy transfer network in the $C_2S_2M_2$-type PSII-LHCII supercomplex, investigating the global dynamics to analyse the possible roles of individual subunits. Here we conducted computational experiments on knockout mutants, by deleting the corresponding sites in the original network. Analysis of the resulting first passage time distributions provides direct insight into the contributions and hence the likely functionality of each component in the overall energy transfer network. In particular, we conclude that (i) The two LHCII trimers are responsible for collecting sunlight and connecting the two monomers. (ii) Minor antennae CP29 and CP26 are the ideal candidates for performing photoprotection, while CP24 is likely only important for structural stabilisation. (iii) Core antennae CP43 and CP47 facilitate both efficiency and photoprotection by balancing the timescales for transfer into reaction centres and transfer out of the PSII core.

We have also shown that the free energy landscape (Fig. 5) that reproduces rates and the equilibrium distribution is relatively flat on the scale of kT at relevant temperatures. There are no significant kinetic traps, and the mean first passage time to a reaction centre from the PSII-LHCII components varies within a range of about four orders of magnitude. This organisation is very different from multifunnel energy landscapes that feature rare events, when the range of relaxation times is typically much greater, because alternative funnels function as kinetic traps. The structure of the energy landscape for PSII-SC supports a wide variety of alternative pathways for energy transfer, which encodes robust functionality and provides the opportunity to tune the balance between photoprotection and energy transfer efficiency under different ambient light conditions. This key design feature corresponds to a high pathway entropy and a relatively flat energy landscape relative to kT.

Our analyses also highlight the importance of resolving distributions in ensemble-averaged kinetics. While single-trajectory measurements are not currently possible in ultrafast spectroscopies, there are methods that could potentially improve our understanding of the heterogeneity of the EET pathways. One possibility is to use exciton-exciton annihilation as a proxy to measure exciton diffusion dynamics. Recently, an intensity-cycling-based method has been proposed to enable easy extraction of $5^{th}$-order responses, which correspond to two-particle dynamics, in transient absorption spectroscopy[71]. This technique facilitates the measurement of exciton-exciton annihilation dynamics. It has been applied to photosynthetic thylakoid membrane to reveal the exciton diffusion length[72]. Applying the same technique to the PSII-SC with different antennae sizes can potentially reveal the pathway heterogeneity. In particular, the heterogeneity of the EET network has been modelled based on exciton diffusion dynamics[21]. While the method treats heterogeneity implicitly, combining experimentally extracted exciton diffusion dynamics and kMC simulations could provide a direct description of pathway heterogeneity and greatly improve the understanding of how it affects the overall EET network and the functions of the PSII-SC. However, directly probing the FPT distributions, particularly those of a single excitation location, still remains challenging and requires further experimental design.

The demands of both efficiency and photoprotection have required PSII to evolve a unique strategy. We have demonstrated that having a relatively flat energy landscape and a multi-component structure allows PSII to successfully adapt to the fluctuating environment. In particular, all subunits have their own role, but also cooperatively contribute to the energy transfer network, providing a systematic way to achieve highly regulated solar energy conversion. This design principle could serve as a paradigm for control mechanisms in artificial solar devices, reaching for the goal of energy sustainability.

## Method

We construct a kinetic model for the EET network in the $C_2S_2M_2$-type PSII-SC based on the method used by Bennett et al.[22] and Leonardo et al.[23]. Briefly, we combine semi-empirical intra-protein Hamiltonians obtained from literature[14,15,73–75] and structure-based inter-protein pigment couplings to build a full Hamiltonian model for the PSII-SC. The cryo-EM structure of the $C_2S_2M_2$-type PSII-SC from *Pisum sativum*[4] (PDB:5XNL) and the TrEsp method[76] are employed for the calculation of inter-protein pigment couplings. Based on the full Hamiltonian, all states are separated into domains according to electronic couplings and degrees of delocalisation (see ref. 22 for details). This definition of domains essentially allows a separation of timescales, i.e. by construction, intra-domain EET is relatively fast and inter-domain EET is relatively slow. It also allows intra-domain EET to be calculated based on modified Redfield theory and inter-domain EET to be calculated with generalised Forster theory. 500 sets of site energies, obtained based on the inhomogeneous broadening widths reported in Table S1, are used for generating 500 inhomogeneous realisations. The averaged rate matrix over all inhomogeneous realisations is used for the kinetic analyses. Future improvement can be done by performing kinetic analyses on individual realisations. However, the analyses will require extensive computing resources. The lowest excitonic state in the RCs is connected to a charge transfer state and the charge separation is assumed to be infinitely fast[21,23]. Parameters used in the kinetic model are described in the SI. We note that energy trapping, both in the RC and at quenching sites in the NPQ simulation, can be improved in the future. However, this extension requires further investigation into the charge transfer states and multiple possible quenching mechanisms, which include energy transfer from Chls to the S1 state of carotenoids or charge transfer between a carotenoid/Chl pair[77]. While a more sophisticated model for charge transfer and energy quenching could lead to a more accurate simulation of energy trapping, the current parameters are chosen empirically to be consistent with experimental observations and/or other simulation results.

Based on the kinetic model for the EET network, we construct the free energy landscape of the $C_2S_2M_2$-type PSII-SC, and investigate the dynamics using kMC simulations[32–34]. This approach provides a way to perform single-trajectory analysis, which captures the inhomogeneity that is hidden in average kinetic behaviour. To systematically characterise the EET network we have analysed the FPT distributions and corresponding kMC trajectories for energy transfer.

We have recently shown how this FPT distribution reports on the underlying structure of the energy landscape, especially the existence of kinetic traps associated with different relaxation timescales[30,68]. Peaks in an appropriate plot of the FPT distribution appear at positions corresponding to the relaxation times sampled from a given initial condition, and can often be assigned to specific features of the underlying landscape[30]. The kinetic transition networks we obtain for the PSII-SC in the present work are relatively small, and a key feature is that they do not feature any rare events. It was therefore possible to apply standard kMC approaches for

individual trajectories along with full eigendecomposition of the transition matrix for FPT analysis.

The free energies of each state are defined to reproduce the equilibrium occupation probabilities, and the free energy barriers reproduce the rates from the underlying kinetic model [Supplementary Information equations (4)-(7)]. These definitions were used to produce the visualisation in Fig. 5.

The eigendecomposition approach to calculate FPT distributions is based on the transition matrix formed from the state-to-state rates in the kinetic model. For every product state or states we set the corresponding escape rates to zero and calculate the eigenvalues and eigenvectors of the resulting substochastic matrix. The first passage time and all its moments can be written in terms of these eigenvalues and eigenvectors for any chosen starting distribution [Supplementary Information equations (9) and (10)].

The dwell time distributions are calculated from the kMC trajectories. Only the trajectories whose FPTs lie within the selected ranges (the shaded areas in Fig. 2c) are counted. For each trajectory, the time spent in each state is summed over the states that belong to each subunit. The integrated time (which can include multiple visits of the same states within a trajectory) is defined as the dwell time of a subunit. The dwell time distributions shown in Fig. 3 are the averaged results from all the chosen kMC trajectories. The mathematical definition of the dwell time distribution is:

$$T_{\text{dwell}}(S, T_{\text{FP}}) = \frac{1}{N_{\text{traj}}} \sum_{t \in T_{\text{FP}}} \sum_{a \in S} T_{\text{dwell}}(a) \qquad (1)$$

where $T_{\text{dwell}}(S, T_{\text{FP}})$ is the dwell time of a subunit $S$ for the trajectories within a certain FPT range $T_{\text{FP}}$, $N_{\text{traj}}$ is the number of trajectories whose FPTs lie within $T_{\text{FP}}$, $a$ is the index of exciton states in the PSII-SC, and $T_{\text{dwell}}(a)$ is the time spent in the exciton state $a$ within a trajectory.

## Data availability
All data presented in this manuscript is available at https://doi.org/10.5281/zenodo.13346121.

## Code availability
The code used to generate the kinetic model and to perform FPT and dwell time analyses are available at https://doi.org/10.5281/zenodo.13346121. PATHSAMPLE and disconnectionDPS are available for use under the Gnu General Public Licence and can be downloaded from www-wales.ch.cam.ac.uk.

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

## Acknowledgements

This research was supported by the US Department of Energy, Office of Science, Basic Energy Sciences, Chemical Sciences, Geosciences, and Biosciences Division. SJY is grateful of the support from the Kavli Energy NanoScience Institute. DJW is grateful to the Miller Institute for a Visiting Miller Professorship, which facilitated the collaborations in this research. EJW gratefully acknowledges support from the Engineering and Physical Sciences Research Council [grant numbers EP/R513180/1, EP/N509620/1]. SJY thanks Prof. Doran I. G. B(ennett) Raccah for sharing the code for simulation and providing useful suggestions.

## Author contributions

G.R.F. and D.J.W. conceived the study design. S.J.Y. constructed the kinetic model. D.J.W. and E.J.W. constructed the free energy landscape and performed kMC simulations. S.J.Y. analysed the simulation results. S.J.Y. wrote the original draft of the manuscript. All authors contributed to the manuscript.

## Competing interests

The authors declare no competing interests.
