## [Peer Review File · Nature Communications]

Design Principles for Energy Transfer in the Photosystem II Supercomplex from Kinetic Transition NetworksReviewer #1 (Remarks to the Author):

The manuscript presents simulations for the global kinetics and energy landscape in the photosystem II supercomplex (C2S2M2). There are many interesting and significant results concerning the kinetics of energy transfer pathways and the role of individual units. Using a dimeric PSII-SC model enables insights into pathways between distinct components of each monomer. The major message concerns the fact that there is a narrow band of closely spaced substates with absence of significant kinetics traps (I appreciated the comparison with other types of system, pages 20-21), creating the conditions for achieving a balance between efficient energy transfer and photoprotection. The work brings together several ideas and is certainly a significant addition to the field. I am not an expert in the specific methodological approach, but it appears sound to me. It would be nice to see however a discussion of the possible limitations of the approach and suggested ways of improving it (not only for the procedure of simulation itself, but also for the primary data, for example, are possible delocalized and CT states considered?). Some conclusions reached in the manuscript have been discussed before in literature that is not cited here. Perhaps not via the same method or on the same system/model, but still closely corresponding to the present work. For example, the attribution of roles to CP26 and CP29 relates to prior works like *Plant Cell Environ.* 2020, 43, 496, *Plant Physiol.* 2023, 193, 1365, *BBA Bioenerg.* 2020, 1861, 148282 (this is not a suggested list for added citations, merely examples to motivate better connection to the literature and clearer explanation of what is novel here - similarly for CP43/CP47). Is there anything to learn by comparison with other types of SC (such as PNAS 2019, 116, 21246)? It would be useful for the more general audience to describe briefly what the molecular nature of the quenchers might be and how they might be activated. Of course the principal message of the manuscript concerns the absence of the kinetic traps and the presence of multiple pathways; here, it would be useful to close with some statement about experimental connections and implications (does the conclusion explain some observations? does it suggest new ways to verify the simulation results?)

Reviewer #2 (Remarks to the Author):

A kinetic model for the EET network for a C2S2M2 PSII-SC was constructed using known methods and based on plant cryoEM structure. From this, a kinetic Monte Carlo simulations was used to obtain large number of single-trajectory results. These were then analysed for the first passage time (FPT) distributions and dwell time distribution. Systematically, using computational “knockoff” and planting quenchers in the different units, the authors can then analyse for the possible functions of the units and where the reasonable sites of NPQ are. These analysis is rather novel and interesting and important conclusions on PSII can be drawn, such as CP29 or M-LHCII (C) are likely quenching sites.

PSII is an biological important system. The analysis is novel and the results regarding the possible sites of quenchers and general design of the various units of PSII is meaningful. The work should appeal to a wide scientific audience. I would recommend acceptance after these issues are addressed:

The methodology lacks details. How were the kinetic Monte Carlo, first passage time, and dwelling time calculated? As the readership is broad, a detailed description of these methods is appreciated. Furthermore, the analyses throughout the manuscript were done on the basis of the results from the modified Redfield--generalized Förster calculation. The author should provide some details, in the SI, on how the energy landscape looks like from the calculation

results?

In the “First Passage Time Analysis” subsection, it is mentioned that more analyses are in the SI. Are the results from them significant? If yes, the authors could briefly describe them in the main text.

There are no comparisons with experiments and hence it is difficult to experimentally validate the results obtained in this study. The authors may want to discuss comparisons of the results that they obtain with published experimental results. Example: Compare the mean FPT (e.g. Figure 2) to fluorescence lifetime in van Amerongen (ref. 17. 2011 Biophysical J) and more recent 2D results by Tan (2024 Sci Adv, albeit in a minimal PSII supercomplex).

In the “Dwelling Time Distribution Analysis” subsection, line 194, if transfer back to RC 1 is partially blocked in koCP26, I would think more FPT will appear in at longer timescales instead, but it doesn't appear so in Fig 2c. Could the author elaborate on this point?

a few more minor points, mostly concerning the SI:

- Just to be clear, the ‘normalized prob’ in Fig 2 b and c means the area under the curve is normalized? It should be stated clearly.
- In Fig S1, what does it mean by “FPT distributions from analytical formulation”? How is the analysis different from kMC?
- Table S1, how about the site energies? Is CP24 included?
- Table S2, “detailed description can be found in the Simulation Details”, but looks like the description is missing.
- Fig 5 caption: not sure what “appear in red” means here.
- Also Fig 5, is it possible to include the name of the subcomponent in each branch, or at least a few representative ones?
- The term RC 1 and RC 2 should be denoted in the figure for better visualization.
- The use of term “substates” and “subcomponents” should be elaborated to avoid confusion.

Reviewer #3 (Remarks to the Author):

Reviewer #4 (Remarks to the Author):

In this study, Yang et al. adapted a structure-based model, originally developed by Bennett et al. in 2013 to calculate energy transfer and trapping within the plant C2S2M2 Photosystem II supercomplex, to analyze the cryo-EM structure as reported by Su et al. in 2017. Initially, they outline potential energy transfer pathways within the complex, subsequently employing the model to pinpoint possible quenching sites. This method of pinpointing the non-photochemical quenching locations was already used when the first structures appeared (see the work of Valkunas et al. PCCP 2009 and Caffarri et al. BJ 2011). The distinctive contribution of this work lies in the utilization of a high-resolution cryo-EM structure. Nonetheless, the findings presented are either not novel or not in agreement with a series of experimental results. Moreover, from a modeling standpoint, the study lacks innovative elements, particularly in how quenching is modeled—overlooking any specific quenching rates, thus rendering the approach somewhat

superficial.

Illustrative examples include:

The authors conclude that PSII possesses a flat energy landscape, a fact well-established since the 1990s, which they also acknowledge in their introduction.

The conclusion that the two types of LHCII trimers are responsible for collecting sunlight and are connecting the monomers in the dimeric supercomplex can already be drawn by looking at the structure and no modeling is needed for that.

Their conclusion that CP26 is important for energy transfer has been shown experimentally not to be correct (van Oort et al. Biophysical Journal, 2010).

The identification of CP29 and CP26 as the most probable quenching sites revisits a once-popular theory, but that has since been disproved by numerous experimental investigations, including studies on mutants (e.g. de Bianchi et al. plant Cell, 2008; de Bianchi et al. Plant Cell, 2011).

Reviewer #5 (Remarks to the Author):

This is an interesting theoretical study investigating the details of the flat free energy landscape of the photosystem 2 supercomplex and how it influences the light-harvesting and photoprotective functions.

An innovative aspect of the present study is the use of analysis tools not appreciated before in our field, namely kinetic Monte Carlo (KMC) simulations to solve the master equations for the populations of exciton states and determine first passage times (FPTs) to the reaction center (RC) for different initial conditions and related dwell times in different subunits. An interesting characterization of the energy transfer is given by the kinetic disconnectivity graph (KDG). At present, I am afraid many readers will find it very hard, if not impossible to appreciate these new tools. The authors face the dilemma that they need to explain these tools to a wide audience and also present the technical details for the experts. Both aspects need improvement as outlined in the following.

As far as I understand, FPTs are obtained by diagonalizing the kinetic matrix or by solving these equations with a KMC approach. Equations for the experts need to be included in the supporting information (SI) to explain how the FPTs are obtained in the two methods. The readers will wonder if there is an advantage of one or the other method. The authors state that KMC allows for an analysis of single trajectories. On the other hand, the distribution functions for the FPTs, obtained with the two methods are not just "consistent" but identical, as Figs. S1 to S4 show. Probably they should be identical, shouldn't they? Can dwell times also be obtained by the diagonalization method or only by KMC?

The FDG (Fig. 5) also needs to be explained in much more detail.

How are the free energies extracted?, from the kinetic matrix?

Equations are needed for the experts and text for the wider audience. Is a vertical line in Fig. 5 an exciton domain, a whole subcomplex or something else? Why does the color of the vertical lines not change in vertical direction? Shouldn't it become easier to cross the free energy barrier to the next state (line) In general, how should one read Fig. 5?

Another point: I (Thomas Renger) am puzzled by the shoulder in the FPT for CP43 at 1 ps. The authors refer to our paper (Raszewski and Renger, 2008) where we found that the transfer from CP43 to the RC is somewhat faster than from CP47 (40 ps as compared to 50 ps) but they should also note that our distribution function (Fig. 13 in Raszewski and Renger) of transfer times from CP43 to RC is narrow with a peak at 40 ps and already zero at 20 ps. Since the present calculations use essentially the same parameters it is unclear why an energy transfer time is obtained that is more than an order of magnitude smaller. Please check the calculation and/or explain.

Thank you for the referees' reports on our manuscript 'Design Principles of the Photosystem II Supercomplex and the Roles of Its Subunits'. We are very grateful to the referees who acknowledge the advances we report, and we have made extensive changes in the manuscript based on their suggestions, which we believe have led to improvements. We have addressed all the comments in detail in the reply below. We hope that the revised manuscript, which includes the additional explanation and discussion suggested by the reviewers, will now be acceptable for publication.

Reviewer #1 (Remarks to the Author):

The manuscript presents simulations for the global kinetics and energy landscape in the photosystem II supercomplex (C2S2M2). There are many interesting and significant results concerning the kinetics of energy transfer pathways and the role of individual units. Using a dimeric PSII-SC model enables insights into pathways between distinct components of each monomer. The major message concerns the fact that there is a narrow band of closely spaced substates with absence of significant kinetics traps (I appreciated the comparison with other types of system, pages 20-21), creating the conditions for achieving a balance between efficient energy transfer and photoprotection. The work brings together several ideas and is certainly a significant addition to the field. I am not an expert in the specific methodological approach, but it appears sound to me. It would be nice to see however a discussion of the possible limitations of the approach and suggested ways of improving it (not only for the procedure of simulation itself, but also for the primary data, for example, are possible delocalized and CT states considered?).

While delocalization of exciton states is included in the current model [ref 22,23], CT states are only included phenomenologically. The following paragraph is included to discuss potential ways to improve the current model, specifically regarding energy trapping. (P26, L585-593)

"We note that energy trapping, both in the RC and at quenching sites in the NPQ simulation, can be improved in the future. However, this extension requires further investigation into the charge transfer states and multiple possible quenching mechanisms, which include energy transfer from Chls to the S1 state of carotenoids or charge transfer between a carotenoid/Chl pair[73]. While a more sophisticated model for charge transfer and energy quenching could lead to a more accurate simulation of energy trapping, the current parameters are chosen empirically to be consistent with experimental observations and/or other simulation results."

Some conclusions reached in the manuscript have been discussed before in literature that is not cited here. Perhaps not via the same method or on the same system/model, but still closely corresponding to the present work. For example, the attribution of roles to CP26 and CP29 relates to prior works like Plant Cell Environ. 2020, 43, 496, Plant Physiol. 2023, 193, 1365, BBA Bioenerg. 2020, 1861, 148282 (this is not a suggested list for added citations, merely examples to motivate better connection to the literature and clearer explanation of what is novel here - similarly for CP43/CP47).

We thank the reviewer for their suggestions. We note that the first two works are not included because they are related to *Chlamydomonas reinhardtii*, which has a different NPQ mechanism because the quenching sites are in LHCSR instead of directly associated with the major or minor antennae. Additional modeling will be required to understand the role of CP26 and CP29 in efficiency and photoprotection in *Chlamydomonas reinhardtii*. The third work the reviewer provided along with other relevant works are now included in the references (ref35-42, 67-68).

Is there anything to learn by comparison with other types of SC (such as PNAS 2019, 116, 21246)?

Comparing kinetic behavior in different forms of the PSII-SC is an ongoing project. This comparison requires construction of EET models for different PSII-SCs, as the models are the basis for the kinetic analyses. The results will be reported in future publications.

It would be useful for the more general audience to describe briefly what the molecular nature of the quenchers might be and how they might be activated. Of course the principal message of the manuscript concerns the absence of the kinetic traps and the presence of multiple pathways;

The following paragraph is included to briefly explain the activation of energy quenching in plants. (P11, L226-P12, L231)

"In plants, excessive illumination will introduce a pH gradient across the thylakoid membrane. This gradient activates a pH-sensing protein and the xanthophyll conversion cycle[35]. The xanthophyll produced under highlight conditions, namely zeaxanthin, can act as a quencher, which accepts excitation energy from Chls and the energy quickly undergoes dissipative pathways[2,3]. To simulate energy quenching, an additional component is added to the rate matrix as a sink."

here, it would be useful to close with some statement about experimental connections and implications (does the conclusion explain some observations? does it suggest new ways to verify the simulation results?)

The following paragraph is added in the conclusion. (P24, L545-P25, L561)

"Our analyses also highlight the importance of resolving distributions in ensemble-averaged kinetics. While single-trajectory measurements are not currently possible in ultrafast spectroscopies, there are methods that could potentially improve our understanding of the heterogeneity of the EET pathways. One possibility is to use exciton-exciton annihilation as a proxy to measure exciton diffusion dynamics. Recently, an intensity cycling based method has been proposed to enable easy extraction of 5th-order responses, which correspond to two-particle dynamics, in

transient absorption spectroscopy[67]. This technique facilitates the measurement of exciton-exciton annihilation dynamics. It has been applied to photosynthetic thylakoid membrane to reveal the exciton diffusion length[68]. Applying the same technique to the PSII-SC with different antennae sizes can potentially reveal the pathway heterogeneity. In particular, the heterogeneity of the EET network has been modeled based on exciton diffusion dynamics[21]. While the method treats heterogeneity implicitly, combining experimentally extracted exciton diffusion dynamics and kMC simulations could provide a direct description of pathway heterogeneity and greatly improve the understanding of how it affects the overall EET network and the functions of the PSII-SC.”

Reviewer #2 (Remarks to the Author):

A kinetic model for the EET network for a C2S2M2 PSII-SC was constructed using known methods and based on plant cryoEM structure. From this, a kinetic Monte Carlo simulations was used to obtain large number of single-trajectory results. These were then analysed for the first passage time (FPT) distributions and dwell time distribution. Systematically, using computational “knockoff” and planting quenchers in the different units, the authors can then analyse for the possible functions of the units and where the reasonable sites of NPQ are. These analysis is rather novel and interesting and important conclusions on PSII can be drawn, such as CP29 or M-LHCII (C) are likely quenching sites.

PSII is an biological important system. The analysis is novel and the results regarding the possible sites of quenchers and general design of the various units of PSII is meaningful. The work should appeal to a wide scientific audience. I would recommend acceptance after these issues are addressed:

The methodology lacks details. How were the kinetic Monte Carlo, first passage time, and dwelling time calculated? As the readership is broad, a detailed description of these methods is appreciated.

Detailed descriptions of the calculations for the first passage time distributions using kinetic Monte Carlo and eigendecomposition methods have been added to the SI, with references to our previous presentations.

The following text has been added to the SI (SI P4, L67-P6, L106):

“Two alternative approaches were employed to calculate the first passage time distribution (FPT), and we summarise them here. First, we explain how eigendecomposition provides an analytical expression for the FPT. We then describe the complementary kinetic Monte Carlo (kMC) method, which samples individual trajectories from source to sink.

We consider the transition matrix $\mathbf{Q} = \mathbf{K} - \mathbf{D}$, where \mathbf{D} is a diagonal matrix of escape rates, with elements $D_{jj} = \sum_{\gamma} K_{\gamma j}$. The kinetics are described by the linear master equation,

$$\frac{d\mathbf{P}(t)}{dt} = \mathbf{Q}\mathbf{P}(t), \quad (1)$$

where $\mathbf{P}(t)$ is the time-dependent vector of occupation probabilities for the substates. We are interested in the first passage time distributions, defined as the first hitting time for a trajectory to reach the sink \mathcal{A} , given an initial starting probability distribution. We set all the escape rates from the sink to zero, and define the substochastic matrix $\mathbf{Q}_S = \mathbf{K}_S - \mathbf{D}_S$, where we have partitioned the state space into two disjoint sets, $\Omega = \mathcal{A} \cup S$. Ω is the full state space, and S is the state space minus the sink. \mathbf{Q}_S is the subset of the full transition matrix \mathbf{Q} containing the interstate transition rates within S . \mathbf{D}_S is the corresponding subset of \mathbf{D} including the escape rates to \mathcal{A} .

Eigendecomposition

The substochastic transition matrix can be decomposed into its constituent eigenmodes

as,

$$\mathbf{Q}_S = - \sum_{\ell}^{|S|} \lambda_{\ell} \mathbf{w}_{\ell}^R \otimes \mathbf{w}_{\ell}^L, \quad (2)$$

where \mathbf{w}_{ℓ}^L and \mathbf{w}_{ℓ}^R are the left and right eigenvectors and \otimes is the outer product. \mathbf{w}_{ℓ}^L is a row vector, and \mathbf{w}_{ℓ}^R is a column vector. All eigenvalues are real and negative, $-\lambda_{\ell} < 0$. Using the above decomposition, we can write the first passage time distribution as a summation over eigenmodes,

$$p(t) = \sum_{\ell=1}^{|S|} \lambda_{\ell} e^{-\lambda_{\ell} t} \mathbf{1}_S (\mathbf{w}_{\ell}^R \otimes \mathbf{w}_{\ell}^L) \mathbf{P}_S(0). \quad (3)$$

Here, $\mathbf{1}_S$ is a row vector of ones and $\mathbf{P}_S(0)$ is the initial occupation probability in S , at $t = 0$. It is useful to make the transformation $y = \ln t$, to produce the probability distribution $\mathcal{P}(y)$,

$$\mathcal{P}(y) = \sum_{\ell=1}^{|S|} \lambda_{\ell} e^{y - \lambda_{\ell} e^y} \mathbf{1}_S (\mathbf{w}_{\ell}^R \otimes \mathbf{w}_{\ell}^L) \mathbf{P}_S(0). \quad (4)$$

As $p(t)$ and $\mathcal{P}(y)$ are normalised distributions, $\sum_{\ell} \mathbf{1}_S (\mathbf{w}_{\ell}^R \otimes \mathbf{w}_{\ell}^L) \mathbf{P}_S(0) = 1$.
Kinetic Monte Carlo

To analyse trajectory information in more detail, we also run kMC simulations, which generate stochastic trajectories starting from the source and terminating at the sink. Standard rejection-free kMC simulations work using two random numbers to generate the next transition and the associated timestep. If the trajectory currently lies in substate i , a random number r_1 is drawn uniformly with $r_1 \in (0, 1]$. The system is progressed to substate j , where,

$$\sum_{k=1}^{j-1} B_{ki} < r_1 \leq \sum_{k=1}^j B_{ki}. \quad (5)$$

The simulation clock time is incremented by $\Delta t = \tau_i \log r_2$, where r_2 is a second random number, also drawn uniformly with $r_2 \in (0, 1]$. \mathbf{B} is the transition probability matrix with elements B_{ij} corresponding to the probability of transferring to substate i given a step is taken out of substate j . This process samples trajectories according to the master equation. Using kMC simulations to compute FPT distributions enables trajectories to be assigned to particular time windows, which facilitates the dwell time distribution analysis."

A further description of the dwell times has been added to the methods section of the manuscript (P27, L611-621):

"The dwell time distributions are calculated from the kMC trajectories. Only the trajectories whose FPTs lie within the selected ranges (the shaded areas in Figure 2c) are counted. For each trajectory, the time spent in each state is summed over the states that belong to each subunit. The integrated time (which can include multiple visits of the same states within a trajectory) is defined as the dwell time of a subunit. The dwell time distributions shown in Figure 3 are the averaged results from all chosen kMC trajectories.

The mathematical definition of the dwell time distribution is:

$$T_{dwell}(S, T_{FP}) = \frac{1}{N_{traj}} \sum_{t \in T_{FP}} \sum_{a \in S} T_{dwell}(a) \quad (6)$$

where $T_{dwell}(S, T_{FP})$ is the dwell time of a subunit S for the trajectories within a certain FPT range T_{FP} , N_{traj} is the number of trajectories whose FPTs lie within T_{FP} , a is the index of exciton states in the PSII-SC, and $T_{dwell}(a)$ is the time spent in the exciton state a within a trajectory.”

Furthermore, the analyses throughout the manuscript were done on the basis of the results from the modified Redfield–generalized Förster calculation. The author should provide some details, in the SI, on how the energy landscape looks like from the calculation results?

We apologize for not being clear. The free energy landscape visualized in Figure 5 is a translation of the transition rate calculation results, which is explicitly stated in Figure 5 now:

“Each branch terminates at one of the exciton states of the PSII-SC, which is the basis for EET calculations.”

The simulation parameters, which include the site energies, are taken from literature, and the sources are listed in the SI.

In the “First Passage Time Analysis” subsection, it is mentioned that more analyses are in the SI. Are the results from them significant? If yes, the authors could briefly describe them in the main text.

The analysis results are in the SI because of the length limitation of the manuscript, as well as to avoid the risk of overwhelming the readers. The results are significant for understanding the energy transfer network within the PSII-SC, which is crucial for discussing the roles of the subunits. While explicit results of the EET network is in the SI, conclusions from these analyses, specifically regarding the role of the subunits, are summarized in Table 1 and discussed in the section “Roles of the PSII-SC Subunits”.

There are no comparisons with experiments and hence it is difficult to experimentally validate the results obtained in this study. The authors may want to discuss comparisons of the results that they obtain with published experimental results. Example: Compare the mean FPT (e.g. Figure 2) to fluorescence lifetime in van Amerongen (ref. 17. 2011 Biophysical J) and more recent 2D results by Tan (2024 Sci Adv, albeit in a minimal PSII supercomplex).

The EET model was subjected to fluorescence lifetime fitting and was in good agreement

with the lifetimes reported in ref 17. However, fluorescence lifetime experiments, which reveals only ensemble averaged dynamics, does not provide microscopic description of the EET pathways and is not enough to completely validate the model. In other words, a similar fluorescence lifetime does not indicate the EET dynamics are the same. In addition, our simulation focuses on single excitation and pathway inhomogeneity. This makes it extremely challenging to directly compare our results with those obtained from experiment because it is currently impossible to achieve single-state excitation and single-trajectory measurements in the PSII-SC. As the FPT distributions (Figure 2b-c and Figure S1-S4) show, different excitation locations lead to very different pathways, and the timescales of the pathways can span up to 4 orders of magnitude. Even by averaging the FPT distribution to obtain the mean FPT, the result is different from fluorescence lifetime because an ensemble excitation (in the case of experiment) will result in a very different timescale than single excitations. Qualitatively, our results also show fast transfer between CP26 and CP43, which is consistent with the observation reported by Tan (2024 Sci Adv). But this should be as far as the comparison goes because, again, the excitation conditions are too different to allow quantitative comparison.

In the “Dwelling Time Distribution Analysis” subsection, line 194, if transfer back to RC 1 is partially blocked in koCP26, I would think more FPT will appear in at longer timescales instead, but it doesn’t appear so in Fig 2c. Could the author elaborate on this point?

The FPT distributions are normalized with respect to the area under the curve. A simple way to think about this is that the number of total trajectories is effectively the same for each FPT distribution. Compared to the WT, more trajectories will go through direct transfer from CP43 to the RC without leaving the core, as shown by the higher probability at short timescales. This scenario leaves fewer trajectories with an FPT longer than about 50 ps for koCP26, and therefore its FPT distribution does not show a higher probability on longer timescales than the WT. However, when comparing the relative distribution between medium timescales (returning back to RC1) and longer timescales (traveling to the D2 side), the WT has a more similar probability for the medium and longer timescales than koCP26, which exhibits moderately higher probability on the longer timescales than medium timescales. This difference means that it is less likely for an excitation to return to RC1 in koCP26 than in the WT, and we conclude that this result is because the removal of CP26 from the energy transfer network partially blocks the transfer pathways back to RC1.

The following sentence is included in the manuscript (P10, L198-201):

“This is also supported by the relatively higher probability at longer FPT (note that the peak is not higher than the probability distribution of the WT due to a significant amount of initial pathways out of the core being blocked and the total area is normalized).”

a few more minor points, mostly concerning the SI:

- Just to be clear, the 'normalized prob' in Fig 2 b and c means the area under the curve is normalized? It should be stated clearly.

The following sentences were added in the figure legend (P6, Fig2) and the main text(P7, L131-133).

"The FPT distributions are normalized according to the area under the curves."

"The analytical FPT distributions are normalized so that the integrated probability over the entire FPT range is unity."

- In Fig S1, what does it mean by "FPT distributions from analytical formulation"? How is the analysis different from kMC?

We have amended the text from "analytical formulation" to "analytical eigendecomposition of the transition matrix", for clarity. We have also added more descriptions of kMC and eigendecomposition to the SI, as mentioned above.

- Table S1, how about the site energies? Is CP24 included?

More details are added in the SI in the revision (SI P2, L21-P3, L40). The details include *"The intra-protein Hamiltonians, including site energies and intra-protein couplings, were obtained from the literature (references are cited in the footnotes of Table S1). Due to the lack of semi-empirical Hamiltonians for CP26 and CP24, in the current PSII-SC model, they are replaced by the Hamiltonians of CP29 and LHCII, respectively, with absent Chls deleted."*

- Table S2, "detailed description can be found in the Simulation Details", but looks like the description is missing.

We apologize for this omission. A detailed description including the relevant equations is now added in the SI. (SI P2, L21-P3, L40)

- Fig 5 caption: not sure what "appear in red" means here.

The branches that are colored in red have the shortest MFPT to the CT state in the RC in the chosen color scheme. These branches correspond to the excitonic states in the RC.

- Also Fig 5, is it possible to include the name of the subcomponent in each branch, or at least a few representative ones?

This was a good suggestion, which we have implemented. Hopefully it also makes the RC states in red clearer.

- The term RC 1 and RC 2 should be denoted in the figure for better visualization.

The specific RCs are labeled now in Figures 1, 2 and 4, as well as the disconnectivity graph in Figure 5.

- The use of term "substates" and "subcomponents" should be elaborated to avoid confusion.

We changed all "subcomponents" to "substates", and specify that the substates are the exciton states in the PSII-SC.

Reviewer #3 (Remarks to the Author):

Reviewer #4 (Remarks to the Author):

In this study, Yang et al. adapted a structure-based model, originally developed by Bennett et al. in 2013 to calculate energy transfer and trapping within the plant C2S2M2 Photosystem II supercomplex, to analyze the cryo-EM structure as reported by Su et al. in 2017. Initially, they outline potential energy transfer pathways within the complex, subsequently employing the model to pinpoint possible quenching sites. This method of pinpointing the non-photochemical quenching locations was already used when the first structures appeared (see the work of Valkunas et al. PCCP 2009 and Caffarri et al. BJ 2011).

In the works the reviewer cites, coarse-grained models are used, and the inhomogeneity of the energy transfer network is either not taken into account or is treated implicitly. That is, the different transfer rates of specific connections and specific pathways are not considered for the simulation of NPQ. In our contribution, we have constructed a model that contains all state-to-state transfer rates. To discuss how different quencher locations can result in different quenching dynamics, it is important to consider how energy transfer is different at each location. This effect is treated explicitly in our kinetic rate matrix.

The distinctive contribution of this work lies in the utilization of a high-resolution cryo-EM structure. Nonetheless, the findings presented are either not novel or not in agreement with a series of experimental results. Moreover, from a modeling standpoint, the study lacks innovative elements, particularly in how quenching is modeled—overlooking any specific quenching rates, thus rendering the approach somewhat superficial.

In our model, we include the microscopic energy flow in contrast to coarse-grained models the references cited above. This is the key for enabling the investigation of pathway inhomogeneity. We agree that only a simple quenching model was used here, but the quenching rates are taken from the literature. We believe the quantitative description of pathway inhomogeneity (including FPT and dwell time analyses) and the energy landscape provides many new aspects of the EET dynamics as well as novel insight into the functional roles of the PSII-SC, which the reviewer appears to ignore.

Illustrative examples include:

The authors conclude that PSII possesses a flat energy landscape, a fact well-established since the 1990s, which they also acknowledge in their introduction.

We stress that we did not *conclude* that the energy landscape is flat. We present a novel quantitative analysis of the energy landscape and connect it directly to the ability of PSII to achieve both efficient energy conversion and photoprotection. The existing description of PSII in terms of a flat energy landscape is not the same as understanding the working mechanism and understanding what 'flat' actually means. Our investigation of kinetic behavior arising from a detailed quantitative characterisation of the energy landscape allows us to understand the key evolutionary design principles by comparison with other types of energy landscapes. In particular, the organisation we have visualised

and analysed is very different from systems that support single functionality via strongly funnelled landscapes. This discovery in itself provides important new insight into PSII.

The conclusion that the two types of LHCII trimers are responsible for collecting sunlight and are connecting the monomers in the dimeric supercomplex can already be drawn by looking at the structure and no modeling is needed for that.

The structure can provide qualitative predictions but our simulation combines structure information and energy landscape, enabling quantitative analyses which demonstrate how important the transfer between the two monomers is. We also present a variety of new results, using the structure information to actually calculate how much time energy spends in each subunit, how strongly different subunits are connected, as well as the actual timescales associated with the transfer. These are all significant advances that the referee has not appreciated.

Their conclusion that CP26 is important for energy transfer has been shown experimentally not to be correct (van Oort et al. *Biophysical Journal*, 2010).

The reviewer cites the work of van Oort et al. which studies the fluorescence lifetime of knockout mutants. Their results show that the fluorescence lifetime of the CP26 knockout mutant and the WT are similar, from which they concluded that “CP26 does not have any direct or indirect effect on the energy transfer and trapping processes.”

First we note that the fluorescence lifetime simply averages over all the information provided by the full distribution. Different FPT distributions can have similar MFPT, but different dynamics/different pathways could be involved. For example, the MFPT of CP43 to RC transfer is 129 ps for the WT and is 107 ps for koCP26. The difference is similar to the difference in the fluorescence lifetime data, but as can be seen in Figure 2c, the FPT distributions are very different. This clearly shows that a similar fluorescence lifetime does not indicate similar dynamical behaviors, which is a point mentioned above (last comment on P6), and the concern also applies to the experiment described by van Oort et al.

Second, the fluorescence lifetime experiment cannot selectively excite a specific Chl. This lack of selectivity means the fluorescence trace is averaged for different excitation conditions. While the absence of CP26 causes excitations in CP43 to have different dynamics (Figure 2c and Figure 3), it does not cause much change for other excitation locations such as CP47. By averaging different excitation conditions the role of CP26 may be obscured. However, by looking at the excitation of CP43 and dwell time distribution analyses, we can see that CP26 does have strong connection with CP43. Exploiting the information in the full FPT clearly provides important new insight here.

Finally, the most important conclusion in the cited work regarding CP26 is perhaps that its absence does not cause a structural change, whereas the absence of other minor antennae changes the form and structure of the PSII-SC. In our study, we focus strictly

on the kinetic effect of removing individual protein subunits. The purpose of this approach is to show how a particular subunit contributes to the overall energy transfer network. We do not take into account the effect of structural changes. In fact, a change in the structure would make it difficult to construct control experiments to demonstrate the kinetic effects of removing an individual protein subunit.

In conclusion, we believe that it may be misleading to draw a conclusion about the EET network in the PSII-SC simply based on the fluorescence lifetime of the knockout mutant.

The identification of CP29 and CP26 as the most probable quenching sites revisits a once-popular theory, but that has since been disproved by numerous experimental investigations, including studies on mutants (e.g. de Bianchi et al. *plant Cell*, 2008; de Bianchi et al. *Plant Cell*, 2011).

The reviewer is entitled to a personal view on where quenching sites are in the PSII-SC, but the debate is far from settled. There are many investigations showing that minor antennae have quenching ability. Here is a list of examples.

1. Avenson et al. *JBC* 2009
2. Holzwarth et al. *Chem. Phys Lett.* 2009
3. Miloslavina et al. *JBC* 2011
4. Ballottari et al. *JPCB* 2013
5. Dall'Osto et al. *Nat. Plants* 2017
6. Mascoli et al. *Chem* 2019
7. Guardini et al. *Nat Plant* 2020
8. Sardar et al. *JCP* 2022
9. Accomasso et al. *Nat Comm* 2024

While the list can be extended, we believe it is clearly sufficient to demonstrate the current status of the debate, which does not coincide with the reviewer's opinion. The challenge for identifying the quenching sites in mutant experiments is the lack of well-defined control experiments, because knocking out minor antennae also introduces changes to the PSII-SC structures. Another difficult challenge is accounting for the heterogeneity of the system. In our contribution, we provide a systematic way to analyze how addition of different sinks to the EET network can result in different outcomes for photoprotection. To clarify the purpose of our analysis, the following paragraph has been added (P17, L358-363).

"We note that there is an ongoing debate on whether CP29 and CP26 actually perform energy quenching.[36-42] Such a discussion is not within the scope of the current work. Rather, we have performed kinetic analysis to show that, due to the different ways the minor antennae participate in the energy transfer network, they can have different abilities and different working mechanisms for maintaining efficiency and performing photoprotection."

Reviewer #5 (Remarks to the Author):

This is an interesting theoretical study investigating the details of the flat free energy landscape of the photosystem 2 supercomplex and how it influences the light-harvesting and photoprotective functions.

An innovative aspect of the present study is the use of analysis tools not appreciated before in our field, namely kinetic Monte Carlo (KMC) simulations to solve the master equations for the populations of exciton states and determine first passage times (FPTs) to the reaction center (RC) for different initial conditions and related dwell times in different subunits. An interesting characterization of the energy transfer is given by the kinetic disconnectivity graph (KDG). At present, I am afraid many readers will find it very hard, if not impossible to appreciate these new tools. The authors face the dilemma that they need to explain these tools to a wide audience and also present the technical details for the experts. Both aspects need improvement as outlined in the following.

To help general readers we have added additional information and details of the methods to the SI, including the computation of first passage time distributions using kinetic Monte Carlo, and eigendecomposition. This discussion can be found in the new SI section 'Computing First Passage Time Distributions'.

We have also extended the explanation of the free energy disconnectivity graph in the main text, to help explain the figure. This section now reads as follows (P20, L434-455):

"This graph provides a visualisation of the effective free energies of the substates, and the transition states that connect them, where the vertical axis corresponds to the free energy. Our construction produces a graph that faithfully reproduces the corresponding rates and equilibrium occupation probabilities (Figure 5). The branches of the graph terminate at effective free energies for the substates of the PSII-SC, and the relative free energies correspond to the equilibrium distribution for the corresponding rate matrix. We join the branches for individual minima when the free energy reaches the threshold corresponding to the highest transition state on the lowest energy path that connects them. Here, the rates between substates are translated into free energies for the effective transition states, so that the entries in the rate matrix are recovered from the corresponding free energy difference, as explained in the SI. Once the branches merge they represent sets of minima that can all interconvert below the free energy threshold defined by the vertical axis. At the top of the graph, all the substates can interconvert, and there is only one branch. If we think about the graph from the top down, the branches representing multiple substates split into disjoint sets when the free energy falls below the highest point on the lowest energy path that connects them. The ordering of substates on the horizontal axis is determined by how the sets split as the free energy decreases, providing a visualisation of the landscape organisation that quantitatively encodes the underlying rate matrix. The spacing between branches that split is chosen to accommodate the branches that will emerge as the free energy decreases further, and the approximate dimeric symmetry of PSII emerges naturally from this construction."

As far as I understand, FPTs are obtained by diagonalizing the kinetic matrix or by solving these equations with a KMC approach. Equations for the experts need to be included in the supporting information (SI) to explain how the FPTs are obtained in the two methods.

Please see the new section ‘Computing First Passage Time Distributions’ in the SI.

The readers will wonder if there is an advantage of one or the other method. The authors state that KMC allows for an analysis of single trajectories. On the other hand, the distribution functions for the FPTs, obtained with the two methods are not just “consistent” but identical, as Figs. S1 to S4 show. Probably they should be identical, shouldn’t they? Can dwell times also be obtained by the diagonalization method or only by KMC?

The two approaches have complementary strengths and weaknesses, and obtaining quantitative agreement between them provides a validation of our analysis. In particular, we are testing the convergence of the numbers obtained from an ensemble of kMC trajectories, and the numerical stability of eigendecomposition, which can be problematical for landscapes featuring rare events. In the absence of numerical issues, eigendecomposition gives the exact analytical result for the FPT distributions. In the limit of infinite kMC trajectories, the first passage time distributions computed using kMC and eigendecomposition are identical. However, as we have a finite set of kMC trajectories, with finite width bins, we describe the distributions as consistent rather than identical, and conclude that our calculations and conclusions are robust. Dwell time distributions for the entire FPT range, can be computed using eigendecomposition. However, we are interested in comparing dwell time distributions for different finite FPT ranges. This time restriction is more easily handled with kMC, because trajectories can be easily assigned to different time windows.

The FDG (Fig. 5) also needs to be explained in much more detail. How are the free energies extracted? from the kinetic matrix? Equations are needed for the experts and text for the wider audience. Is a vertical line in Fig. 5 an exciton domain, a whole subcomplex or something else? Why does the color of the vertical lines not change in vertical direction? Shouldn’t it become easier to cross the free energy barrier to the next state (line) In general, how should one read Fig. 5?

We have added an explanation of how free energies are generated from rates in the SI. The additional SI text is:

“To visualise the free energy landscape, we translate the rate matrix \mathbf{K} , into effective free energies. Each element K_{ij} is the transition rate from substate j to substate i . The effective free energies $f_s(T)$, for each state s , are defined in terms of the equilibrium occupation probabilities, π_s ,

$$f_s(T) = -k_B T \ln \pi_s, \quad (7)$$

where k_B is Boltzman's constant and T is the temperature. The effective free energy of the transition state that connects substate s to substate s' is $f_{ss'}^\ddagger(T)$, which is chosen to

reproduce the rate constants via the Eyring–Polanyi equation:

$$K_{s's} = \frac{k_B T}{h} \exp \left[-\frac{\left(f_{ss'}^\dagger(T) - f_s(T) \right)}{k_B T} \right], \quad (8)$$

where h is Planck's constant. Rearranging gives,

$$\begin{aligned} f_{ss'}^\dagger(T) &= f_{s'}(T) - k_B T \ln K_{ss'} + k_B T \ln(k_B T/h), & (9) \\ &= f_s(T) - k_B T \ln K_{s's} + k_B T \ln(k_B T/h). & (10) \end{aligned}$$

The $f_s(T)$ values were obtained by exploiting the detailed balance relations defined by the rate matrix entries and minimising a least squares problem using the GMIN global optimisation program."

We have also extended the disconnectivity graph description in the main text, to help explain the figure. The branches terminating at single minima are now colored according to the MFPT for that substate to clarify the representation, and we have added a note to the figure caption to explain this scheme. Branches corresponding to multiple minima are grey. The disconnectivity graph description now reads as follows (P20, L434-455):

"This graph enables the visualisation of the effective free energies of the substates, and the transition states that connect them. This construction produces a graph that faithfully reproduces the corresponding rates and equilibrium occupation probabilities (Figure 5). The vertical axis corresponds to the free energy. The branches of the graph terminate at effective free energies of the substates of the PSII-SC. These substates are grouped into superbins at discrete, regularly spaced, energy thresholds, E , which correspond to the energy levels at which branches join together in the disconnectivity plot. For each discrete energy E , minima lie within the same superbins if the highest transition state energy on the minimum energy pathway between the minima, is lower than E . Therefore, the branches merge at the lowest free energy threshold where substates can interconvert, which is determined by the corresponding rates. The arrangement of states on the horizontal axis is chosen to highlight the structure of the graph, so that branches do not cross and the approximate dimeric symmetry is apparent."

Another point: I (Thomas Renger) am puzzled by the shoulder in the FPT for CP43 at 1 ps. The authors refer to our paper (Raszewski and Renger, 2008) where we found that the transfer from CP43 to the RC is somewhat faster than from CP47 (40 ps as compared to 50 ps) but they should also note that our distribution function (Fig. 13 in Raszewski and Renger) of transfer times from CP43 to RC is narrow with a peak at 40 ps and already zero at 20 ps. Since the present calculations use essentially the same parameters it is unclear why an energy transfer time is obtained that is more than an order of magnitude smaller. Please check the calculation and/or explain.

We note that Figure 13 of [Raszewski and Renger JACS, 2008] shows the distribution function that originates from disorder in site energies. Different combinations of site

energies lead to different compartment transfer rates (eq 35 in Raszewski and Renger), which causes the distribution. In our calculation, the FPT is a distribution because there is pathway inhomogeneity, not because of static disorder. The pathway inhomogeneity arises from the fact that an excitation can go through different pathways even when the initial states and the final states are fixed. For example, direct transfer from CP43 to the RC should contribute to short FPTs while excitation travelling to the periphery before entering the RC should contribute to longer FPTs. This information is not directly revealed in the compartment transfer rate distribution function.

Following Raszewski and Renger we obtain the distribution function below from our model. It has very similar features to the one reported in Raszewski and Renger, narrow and already zero at 10 ps. The disorder-averaged transfer rate for CP43→RC in our model is $(25\text{ps})^{-1}$, which is faster than the $(41\text{ps})^{-1}$ reported in Raszewski and Renger, but slower compared to the $(17\text{ps})^{-1}$ reported in [Bennett et al. JACS 2013] (SI Figure S2). Such a difference most likely originates from the different structures used in the 3 models.

Figure 1: (a) Probability distribution of time constants for transfer from CP43 to RC at 300 K obtained from our kinetic model. (b) Figure 13 from [Raszewski and Renger JACS 2008]. The red distribution in the left panel is the probability distribution of time constants for transfer from CP43 to RC at 300 K.

Reviewer #1 (Remarks to the Author):

All points and concerns have been addressed in the revised version of the manuscript.

Reviewer #2 (Remarks to the Author):

We think the description of the methods are presented much more clearly now. The other issues are also by and large resolved. We therefore recommend acceptance. Here are some minor points on presentation:

Figure 5 seems to be a deep description of PSII-SC energy landscape and EET network.

However, due to the large amount of information it contains, I think the figure will confuse many readers. Perhaps the authors may want to consider move Figure 5 and its description to the SI, while keeping the discussion to the main text.

Another point regarding Figure 5. The figure labels a few branches with the name of the protein complex it belongs to. This can be misleading, since it looks like each protein complex is represented by only one branch. If I understand it right, each branch is an exciton state and thus, a protein complex should correspond to multiple branches, not one. To avoid ambiguity is there a better way to label them, instead of using for example "CP43" for one single branch?

Reviewer #3 (Remarks to the Author):

Reviewer #5 (Remarks to the Author):

The authors have improved the manuscript in all points except one:

I am even more puzzled now by the shoulder in the distribution of FPT for CP43 at 1 ps, which is not seen in the distribution function of the energy transfer time constants (Raszewski and Renger)

CP43->RC, which only contains much longer times.

In their response letter, the authors provide the following reasoning:

1) The distribution function of FPTs does not reflect static disorder, but only pathway inhomogeneity.

My concern: How was static disorder included? In Table S1, values for the widths of the site energy distribution functions of the different subunits are provided. Where these values used in the calculation of FPTs or not? If not, which realization of static disorder was taken? How representative is it?

2) The distribution function of FPTs reflects different pathways, direct transfer from CP43 to RC

should contribute to short FPTs, whereas longer pathways involving the peripheral complexes contribute to longer FPTs.

My concern: In the calculations of the distribution function of energy transfer time constants (Raszewski and Renger)

CP43→RC, only the direct pathway CP43 to RC was considered. So, how can an inclusion of detours of the excitation energy via

peripheral complexes shorten the direct transfer time by one order of magnitude?

In other words, if one needs 6 hours to drive from Los Angeles to San Francisco, why should it take 6 minutes

if one takes into account the possibility to take a detour via New York? What aspect am I missing?

3) The authors have calculated the distribution function of energy transfer from CP43 to the RC, taking into account static disorder and arrive at a very reasonable result (Fig. 1 (a) of their response letter).

My concern: I am relieved to see that this result is in the same order of magnitude as previous results

(Raszewski and Renger, Benneth et al.). However, I am still puzzled how this distribution function

relates to the FFPTs of CP43. I could understand that FFPTs contain longer time constants because of

the detours of excitation energy via the peripheral complexes, but why do they provide one order of magnitude

faster time constants? I strongly recommend to include Fig. 1(a) into the SI and explain in detail how it is related to FPTs of CP43. If these differences are explained, the authors should use the opportunity to

point out which type of experiments can be used to detect one or the other distribution function.

Reviewer #1 (Remarks to the Author):

All points and concerns have been addressed in the revised version of the manuscript.

Reviewer #2 (Remarks to the Author):

We think the description of the methods are presented much more clearly now. The other issues are also by and large resolved. We therefore recommend acceptance. Here are some minor points on presentation:

Figure 5 seems to be a deep description of PSII-SC energy landscape and EET network. However, due to the large amount of information it contains, I think the figure will confuse many readers. Perhaps the authors may want to consider move Figure 5 and its description to the SI, while keeping the discussion to the main text.

The graph in Figure 5 provides a faithful representation of the free energy landscape, which can be translated directly into the equilibrium occupation probabilities of the states and the interconversion rates between them. Disconnectivity graphs have proved to be very valuable in understanding the properties of both molecular and condensed matter systems, and the structure revealed in Figure 5 provides a direct and quantitative view of a "flat" landscape. These graphs now appear throughout the chemical physics literature and in textbooks (e.g. "Energy Landscapes: With Applications to Clusters, Biomolecules and Glasses." D. Wales Cambridge University press ISBN 0-521-814157-4 (2003), and refs 26-31, 50-62, and 66-67.). We believe that this tool will be equally valuable for the photosynthesis community, and we would like to retain the figure in the main text.

Another point regarding Figure 5. The figure labels a few branches with the name of the protein complex it belongs to. This can be misleading, since it looks like each protein complex is represented by only one branch. If I understand it right, each branch is an exciton state and thus, a protein complex should correspond to multiple branches, not one. To avoid ambiguity is there a better way to label them, instead of using for example "CP43" for one single branch?

We added the following sentence in the figure caption to improve the clarity. (Figure 5)

"The initial excitation locations (CP43, CP47, S-B, and M-B) and the final states (RC1 and RC2) discussed in the Results section are pointed out by the arrows and labelled accordingly."

We also added text in the figure to directly reflect that each branch corresponds to an exciton state of the PSII-SC.

Reviewer #3 (Remarks to the Author):

Reviewer #4:

Based on the request from the editor, the following paragraph is included in the main text. (P17, L359–365)

“The analysis of mutant data in studies designed to answer this question is complex, as it is clear that deletion of, minor complexes significantly perturbs the structure of the PSII-SC [19,37,39,43,44]. In contrast, the results and analysis in this work refer to the roles of these complexes within an intact $C_2S_2M_2$ -type PSII-SC, a perspective that is not available from mutant studies. We are not able to comment on the roles of CP26 and CP29 that are detached from the PSII-SC [45].”

Reviewer #5 (Remarks to the Author):

The authors have improved the manuscript in all points except one: I am even more puzzled now by the shoulder in the distribution of FPT for CP43 at 1 ps, which is not seen in the distribution function of the energy transfer time constants (Raszewski and Renger) CP43 → RC, which only contains much longer times. In their response letter, the authors provide the following reasoning:

(1) The distribution function of FPTs does not reflect static disorder, but only pathway inhomogeneity.

My concern: How was static disorder included? In Table S1, values for the widths of the site energy distribution functions of the different subunits are provided. Where these values used in the calculation of FPTs or not? If not, which realization of static disorder was taken? How representative is it?

Static disorder is included in the rate matrices. 500 inhomogeneous realizations are obtained by adding to the site energies a random variable that samples a Gaussian distribution with a zero mean and a standard deviation whose value is reported in Table S1. The averaged rate matrix is then used for the kinetic analyses. Ideally, these analyses should be done on an individual rate matrix from all inhomogeneous realizations. However, this requires extensive computing resources as the number of trajectories required to reproduce results from eigendecomposition of each individual rate matrix is the same as that for the averaged rate matrix. Future improvement can be done by performing kinetic analyses on each inhomogeneous realization.

The following paragraph is included in Methods. (P26, L591–596)

“500 sets of site energies, obtained based on the inhomogeneous broadening widths reported in Table S1, are used for generating 500 inhomogeneous realizations. The averaged rate matrix over all inhomogeneous realizations is used for the kinetic analyses. Future improvement can be done by performing kinetic analyses on individual realizations. However, the analyses will require extensive computing resources.”

2) The distribution function of FPTs reflects different pathways, direct transfer from CP43 to RC should contribute to short FPTs, whereas longer pathways involving the peripheral

complexes contribute to longer FPTs.

My concern: In the calculations of the distribution function of energy transfer time constants (Raszewski and Renger) CP43 → RC, only the direct pathway CP43 to RC was considered. So, how can an inclusion of detours of the excitation energy via peripheral complexes shorten the direct transfer time by one order of magnitude? In other words, if one needs 6 hours to drive from Los Angeles to San Francisco, why should it take 6 minutes if one takes into account the possibility to take a detour via New York? What aspect am I missing?

We believe the distribution at short FPTs will be observed in the model of Raszewski and Renger as well if the FPT distribution is calculated based on their model. First, the disorder-averaged timescales (25 ps in our model, mentioned in the previous response) and the 40 ps timescale in the Raszewski and Renger model are the **thermally and inhomogeneously** averaged transfer times. That is, they are obtained by averaging over two distributions—states in the domains and inhomogeneous realizations (static disorder). In our model, the fastest transfer rate from a CP43 state to the RC in the inhomogeneously averaged transfer rate matrix (without thermally averaging over states) is 7.7 ps. This shows that there are microscopic pathways faster than the thermally averaged timescale 25 ps, which will show up in the FPT distribution. Second, transfer will happen at much earlier times than the time constant. Therefore, it is reasonable that the FPT distribution of CP43 to RC transfer has a shoulder around 2-3 ps.

The dwell time analysis of the shoulder on the short FPT for initial excitation in CP43 shows that only CP43 and the RC are involved (not shown in the main text), which indicates that only direct transfer occurs on this timescale.

3) The authors have calculated the distribution function of energy transfer from CP43 to the RC, taking into account static disorder and arrive at a very reasonable result (Fig. 1 (a) of their response letter).

My concern: I am relieved to see that this result is in the same order of magnitude as previous results (Raszewski and Renger, Bennet[t] et al.). However, I am still puzzled how this distribution function relates to the FFPTs of CP43. I could understand that FFPTs contain longer time constants because of the detours of excitation energy via the peripheral complexes, but why do they provide one order of magnitude faster time constants? I strongly recommend to include Fig. 1(a) into the SI and explain in detail how it is related to FPTs of CP43. If these differences are explained, the authors should use the opportunity to point out which type of experiments can be used to detect one or the other distribution function.

We apologize for not being clear on this point. We consider that the fundamental distinction between our FPT distributions and the inhomogeneous distribution of the thermally averaged rate constants of Raszewski and Renger lies in the relationship of these rate constants to the individual trajectories calculated in our work. As there are multiple microscopic pathways that the excitation can take, individual trajectories can be significantly shorter than the average. Of course, it is also true that some events take place on shorter timescales than the rate constant and these are emphasized by the

logarithmic time axis of the FPT plots. Taking both of these points together leads us to believe that no serious contradiction exists between the work of Raszewski and Renger and our results.

The distribution function in Raszewski and Renger is a distribution of thermally averaged rates, i.e., a single rate in the distribution function still has the averaged nature. For each rate in the distribution, an FPT distribution can be obtained. We believe the FPT distribution for the components in the static disorder distribution function of Raszewski and Renger will show a distribution of short FPTs, as demonstrated here and in Figure 2 in the main text.

Currently, there are no experimental techniques for single trajectory measurements of energy transfer dynamics, making it difficult to measure the FPT distribution. However, single molecule experiments have been demonstrated to be able to show the disorder distribution function.

We took the reviewer's advice and include Figure 1(a) from the previous response in the SI. The following text is also included. (P21, L268–P23,L288)

“Figure 2b-c in the main text show distribution at short FPTs (on \sim ps timescales). This is seemingly contradictory with the CP43 to RC transfer reported in literature, which is typically in tens of ps timescales (40 ps in Ref [4] and 17 ps in Ref [1]). We note that these values reported in literature are the mean transfer rates from CP43 to the RC, averaged based on thermal population of the states within a domain/compartiment. Particularly, Figure 13 in Ref [4] shows the distribution of these averaged rates over different inhomogeneous realizations (static disorder). This distribution shows how static disorder influences the transfer between CP43 to the RC, which is fundamentally different from the FPT distribution discussed in this work. The FPT distribution shows the different microscopic pathways present in the complex transfer network, as illustrated in Figure 2a in the main text. In the case of CP43 to RC transfer, the FPT distribution shows that there are pathways much faster than the averaged rate. To compare with literature, we plot the distribution function according to the definition in the work of Raszewski and Renger [4]. Figure S9 shows the static-disordered rate distribution obtained from our model, which is similar to the one reported in Figure 13 of Ref [4]. Our model gives an inhomogeneously averaged transfer rate of $(25\text{ps})^{-1}$ compared to $(40\text{ps})^{-1}$ reported in Ref [4] and $(17\text{ps})^{-1}$ reported in Ref [1]. These values are of similar magnitude, and the small variations most likely originate from the different structures used in each model. In conclusion, no serious contradiction exists between our work and the results reported in the literature.”

Reviewer #5 (Remarks to the Author):

The authors added a new subsection "Fast Transfer Pathways from CP43 to the RC" in the SI, where they present the probability distribution of the CP43 to RC transfer times (thermally averaged over the domain populations), which is in good qualitative agreement with previous work from the literature. The authors stated in their letter, that it will be very difficult to measure the short transfer times appearing in their probability distribution of first passage times (FPTs) from CP43 to the RC (Fig. 2b of the main text).

I think this is an important message not just for the reviewer but for the readers of this work that might even stimulate them to try to perform new experiments to resolve this transfer. It looks like, the authors forgot to change their discussion of these short transfer times in the main text, which still reads (without reference to the SI, I hope I did not overlook something): "This difference is direct evidence of faster EET from CP43 to the RC than from CP47. This phenomenon has been discussed by Raszewski and Renger [14], who proposed that the difference results from the location of the lower energy states, which are closer to the RC in CP43 than in CP47." This statement is true, but a clarification of the different time scales obtained in [14] (and the present Fig. S9, which should be discussed in the main text) as compared to Fig. 2b is needed. Also, it should be clarified that the short times of the present FPTs are hard to measure in an ensemble experiment but will require more sophisticated approaches (single molecule studies?).

Is this a principal problem or could one hope to find these short times as fast components of a dispersive kinetics of the ensemble?

Reviewer #5 (Remarks to the Author):

The authors added a new subsection "Fast Transfer Pathways from CP43 to the RC" in the SI, where they present the probability distribution of the CP43 to RC transfer times (thermally averaged over the domain populations), which is in good qualitative agreement with previous work from the literature. The authors stated in their letter, that it will be very difficult to measure the short transfer times appearing in their probability distribution of first passage times (FPTs) from CP43 to the RC (Fig. 2b of the main text). I think this is an important message not just for the reviewer but for the readers of this work that might even stimulate them to try to perform new experiments to resolve this transfer.

In the previous response, we mentioned that there are currently no experimental techniques for single-trajectory measurements, which makes it difficult to experimentally construct the FPT distribution and hence obtain the short timescales for the CP43 to RC transfer. Such a concept is already discussed in the main text (P25, L575-591). In the current revision, the following sentence is included to explicitly mention the difficulty of measuring the FPT distribution. (P26, L591-592)

"However, directly probing the FPT distributions, particularly those of a single excitation location, still remains challenging and requires further experimental design."

It looks like, the authors forgot to change their discussion of these short transfer times in the main text, which still reads (without reference to the SI, I hope I did not overlook something): "This difference is direct evidence of faster EET from CP43 to the RC than from CP47. This phenomenon has been discussed by Raszewski and Renger [14], who proposed that the difference results from the location of the lower energy states, which are closer to the RC in CP43 than in CP47." This statement is true, but a clarification of the different time scales obtained in [14] (and the present Fig. S9, which should be discussed in the main text) as compared to Fig. 2b is needed.

We apologize for overlooking the change to the main text. We now include a short paragraph to address the difference in the main text and refer the reader to the SI for a detailed comparison between our results and those from ref [14]. (P7, L157-P8, L163)

"It is important to note that in Ref [14], the distribution function does not show the short transfer timescale (~1 ps) between CP43 and the RC. This is because the distribution function in Ref [14] shows the thermally averaged timescales of different inhomogeneous realizations, whereas the FPT distribution shows timescales

involved in different pathways. The difference arises from the fact that the averaged timescales do not reflect individual pathways. In fact, there is no major discrepancy between our results and those reported in Ref [14], which is discussed in more detail in the SI.”

Also, it should be clarified that the short times of the present FPTs are hard to measure in an ensemble experiment but will require more sophisticated approaches (single molecule studies?). Is this a principal problem or could one hope to find these short times as fast components of a dispersive kinetics of the ensemble?

At this moment, this is more of a principal problem. The FPT distribution shows the inhomogeneity of pathways, not the inhomogeneity of the protein structure. Even within the same protein structure, i.e., with a defined Hamiltonian, there is still a distribution of pathways, which cannot be easily disentangled even in single molecule studies. As mentioned in the first response, the paragraph on P25 (L575-592) addresses the difficulty of experimentally probing the FPT distribution.